# Vision and Site: Revisiting a Pure Land Cave of Dunhuang

Zhenru Zhou and Luke Li *

School of Architecture, Tsinghua University, Beijing 100084, China; zhenru@zhenruzhou.com
* Correspondence: liluke@mail.tsinghua.edu.cn

**Abstract:** Buddhist Utopian vision shaped the art of Pure Land; so did many other factors, including the actual locale. Taking Mogao Cave 172 as the main case study, this article deciphers a visual paradigm of a Pure Land painting and cave in Dunhuang (Gansu, China) from the high Tang period (710–780 CE). By analyzing the visual contents and compositions, the painting medium, the cave spaces, and the cliff site, this study investigates the ways in which the architectural images and spaces in Cave 172 helped to convey the invitation to Pure Land. A close reading of the Western Pure Land painting in Cave 172 reveals the spatial construct of the Buddhist paradise that encouraged a transformative viewing experience. A situated visual analysis of Cave 172 with its auxiliary cave and neighboring caves illustrates the historical procedure in which Pure Land imageries were further integrated with the architectural spaces of caves and cave suites. As this study demonstrates, strategies of spatial layering, self-symmetry and scaling, and plastic and multimedia practices of cave-making enhanced the situatedness of the utopian vision.

**Keywords:** Mogao Cave 172; Meditation Sūtra transformation tableau; architectural painting; Tang dynasty; cave grouping; open-air mural; timber-framed façade; utopian vision; situatedness

## 1. Introduction

Utopian vision characterizes the art of Pure Land, a major genre of East Asian Buddhist art. The Pure Land (Chn: *Jingtu* 淨土, Jpn: *Jodo*), a Buddhist paradise, denotes a set of ideas and practices based on world systems other than our own embodied earthly realms (Eltschinger 2020). For Pure Land adherents, Pure Land art is a visual aid for perceiving the possibility of rebirth in the supremely blissful buddha-fields, among which the Western Pure Land of Amitābha, the buddha of limitless life, is one of the most desirable (Wang 2003).[1] Palatial buildings rising from lotus ponds distinguish the topography of the Western Pure Land. Emerging around the fifth century in China, this imagery has been conveyed through sculptural and pictorial mediums and spatial installations and has circulated across regions of varied climates and topographies in East Asia. The largest known repository of Pure Land paintings of the Tang dynasty (618–907 CE) is the Mogao Caves near Dunhuang (present-day Gansu province), a major Buddhist cave site in the Gobi Desert of Northwest China. The Mogao complex consists of nearly 500 decorated caves, and nearly 100 of them display the theme of Pure Lands. During the High Tang period (710–80), scenes of the Western Pure Land were painted 25 times at the Mogao caves, becoming the most depicted subject matter and impacting the subsequent development of cave designs in Dunhuang (Wang 2001, pp. 15–16).[2] In a few instances like Mogao Cave 172, the cave space was even encompassed by two or more large Pure Land paintings, producing an immersive visual stimulation of the ideal Buddhist paradise.

The visual paradigm is hardly constrained by locale and time. The pictorial mediums and the embodied viewing of the Pure Land are, however, conditioned by the actual space and site. Thus, Dunhuang Pure Land caves, while containing idealized images, are part of the larger histories from which they emerged. Mogao Cave 172, a representative Pure Land cave of the High Tang period, has enhanced our knowledge of Tang-period temple

rituals and artistic competition (Wu 1992a), monastic architecture and architectural painting (Xiao 1989, pp. 70–72, 258–62; Ho 1992, pp. 171–73; Sun and Sun 2001, p. 111), layouts of water landscape and building complexes (Liu 2009; Zhang 2019), and so forth.

Building upon these studies, the current article further asks how the architectural images and spaces in Cave 172 helped to convey the invitation to Pure Land. It not only investigates the image of Pure Land paintings, but also the painting medium and the site. The concept of site expands from the cave chamber the mural decorates to the cave suite the chamber constitutes, and the cliff section the cave suite belongs to. A close reading of the Pure Land painting in Cave 172 reveals how the Buddhist utopia was spatially constructed and made accessible to viewers. A situated visual analysis of Cave 172 with its auxiliary cave and neighboring caves illustrates how Pure Land imagery can be integrated with the architectural spaces of caves and cave suites. By exploring the practice of constructing a Pure Land at Mogao, this study aspires to shed light on the situated-ness of the utopian vision.

## 2. Pictorial Image of the Pure Land

The images of the Western Pure Land inside the Dunhuang caves are mostly sūtra paintings, or what specialists would call a "transformation tableaux" (*bianxiang* 變相).[3] They are pictorial renditions related to three principal scriptures of Pure Land Buddhism and act as a visual aid for contemplating the Buddhist sacred realms. While none of the hundred or more Pure Land paintings from the Mogao caves are identical, the water pond and palatial architecture characterize the imaginary topography. A brief overview of the developments of these visual elements in the Dunhuang murals will illustrate a rich visual paradigm. Afterwards, this section will closely examine the architectural representation in a sūtra painting in Mogao Cave 172, revealing its correspondence with figural images and the sequential way of meditation.

### 2.1. Brief Overview of Pure Land Topography in Dunhuang Murals

The lotus pond is a basic topographical element throughout Pure Land paintings, whereas courtyard complexes have gradually been developed since the Tang. Since the lotus pond was believed to be a threshold of the Western Pure Land, early depictions centered on water imagery (Wong 1998/1999, p. 67; Feng 2018, pp. 196–201). An early painting of the Western Pure Land in Mogao Cave 393 from the Sui period (581–618 CE) depicts minimal environmental elements, namely, a pond of irregular shape below the lotus thrones of the Amitābha triad (Figure 1a).

Architectural elements appeared and multiplied in the Tang, partly addressing the descriptions of ornate land, terraces, and pavilions in Pure Land scriptures, and partly reflecting the developments of architecture in real life. The early Tang period (618–705) saw the emergence of large platforms and rectangular ponds. At the turn of the seventh and eighth centuries, when single buildings began to acquire as much architectonic detail as the later paintings did, the foreground was still designed relatively simply—usually a platform separated from the main platform by a stripe of a lotus pond (Figure 1b). From the High Tang period, the Pure Land paintings began to acquire a composition of multiple terraces and bridges in the foreground. The established composition was also applied to visual representations of other buddha lands such as the Eastern Pure Land of Medicine Buddha (Skt: *Bhaiṣajyaguru*). The mid-Tang period (781–848), alternatively known as the Tibetan period, saw a sharp increase in architectural elements along the central axis (Figure 1c). Sometimes, the entrance hall, colonnade, and corner towers are represented in the foreground, completing the layout of the courtyard complex. The imagery continued to flourish at the Mogao caves in the Guiyijun period (851–1036), often adorned by more ornamental pavilions (Figure 1d) (Shi 1999, p. 20; Wang 2013, pp. 82–83).

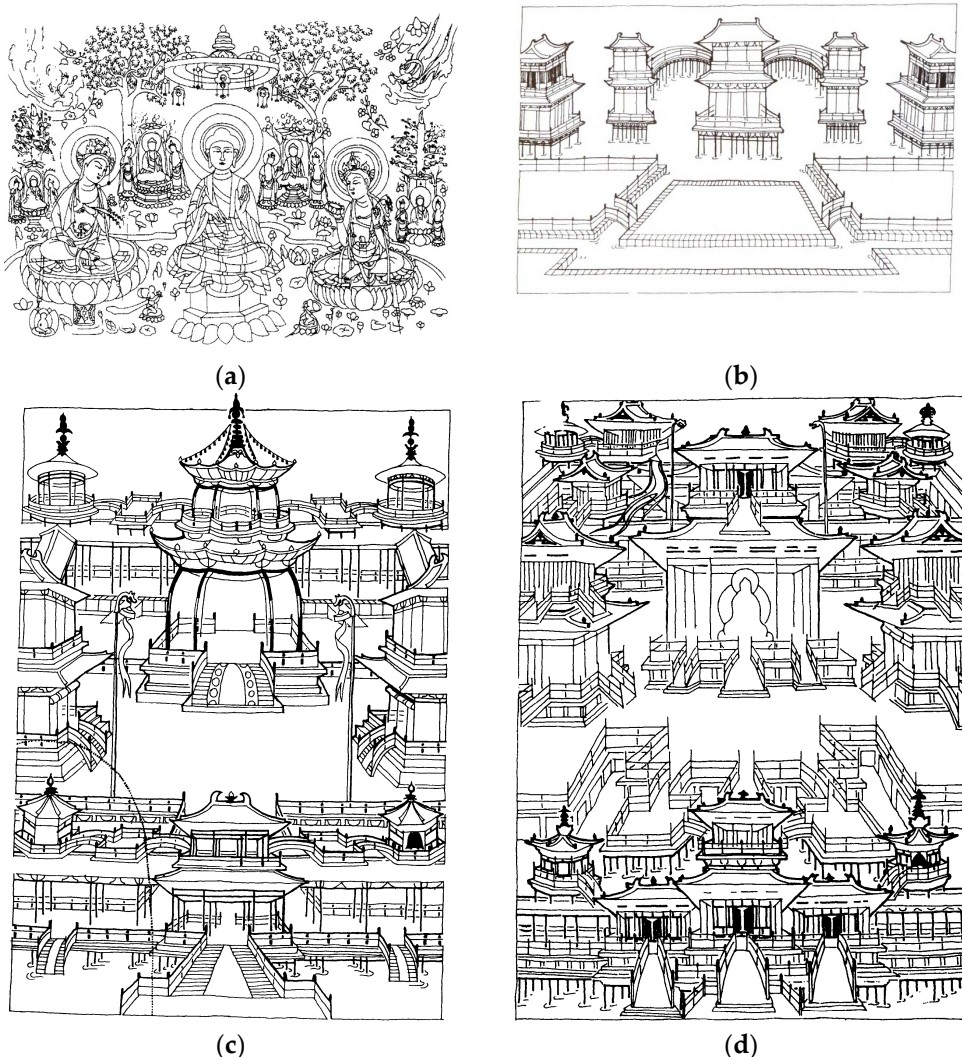

**Figure 1.** Representative pictorial compositions of the Pure Land transformation tableaux in Dunhuang between the sixth and twelfth centuries. Line drawings. (**a**) Western Pure Land, west wall, Mogao Cave 393, Sui period; (**b**) Western Pure Land, north wall, Mogao Cave 205, the early Tang period; (**c**) Eastern Pure Land, north wall, Mogao Cave 361, mid-Tang period; (**d**) Eastern Pure Land, north wall, Mogao Cave 146, Five Dynasties period. After Shi (1999, p. 20); Xiao (1989, pp. 65, 73, 77, figs. 28, 36, 40).

The pictorial composition is flexible for alterations in detail, but the visual logic of a transformative journey laid out in an architectural space continued to thrive from the beginning of the Tang period. Dunhuang mural painting reflects a visual paradigm that was initiated at the cultural centers of Tang China, such as Chang'an in the seventh century, and then circulated to peripheral areas such as Dunhuang. As architectural historians Xiao Mo and Puay-peng Ho suggest, the idealized palatial complex in the Pure Land image was modeled after prototypes in real life, such as urban Buddhist monasteries and imperial palaces in Tang capital cities (Xiao 1989, pp. 61–63; Ho 1995). In addition, numerous studies have acknowledged the Tang Empire's influence on Dunhuang art (Whitfield et al. 2015, p. 73). A recent study by art historian Anne N. Feng takes the Pure Land image as "a symbolic form of Tang opulence and prosperity" (Feng 2018, pp. 1–2). After the An Lushan Rebellion in 755, Dunhuang was successively ruled by the Tibetans, the Guiyijun regime, and a few other powers in the northwest (Rong 2013, pp. 37–49). Therefore, the exchanges between Dunhuang and central China were not as strong as before and local tastes were pronounced. The subsequent development of Pure Land painting in Dun-

huang was largely grounded on the Tang template and filled in more details that may or may not have been applicable in real construction (Bulling 1955, pp. 120–21). Therefore, it is generally accepted that the Pure Land topography in its maturity, as well as architectural painting, is epitomized by mural paintings made in the High Tang period.

### 2.2. *The Central Scene of the Cave 172 Painting*

One of the most frequently cited pictures of the Western Pure Land is a transformation tableau in Mogao Cave 172 (Figure 2).[4] The two side walls of this east-facing, square-planned cave are covered by two mural paintings of an identical subject matter and similar pictorial compositions. Both paintings are based on a Pure Land Buddhist scripture titled *Sūtra of the Meditation on the Buddha of Immeasurable Life* (Foshuo guan wuliangshou fo jing 佛說觀無量壽佛經, hereafter the *Meditation Sūtra*) (*T* no. 365, vol. 12), translated by Kālayaśas 畺良耶舍 (383–442) between 424 and 442 CE. In addition, both paintings feature a tripartite composition comprising a central panel depicting Amitābha's paradise and two side panels of the Ajātaśatru narrative and the sixteen meditations. The pictorial narrative, on the west side, illustrates the circumstance in which the Buddha spoke of the sūtra. The sixteen meditations, on the east side, serves as a visual guide for meditation. Due to its visual pre-eminence and confrontational representation, it is the central scene that absorbs the beholder's attention. While mirroring each other in terms of general composition, the two Pure Land tableaux have distinctive painting styles and nuances in visual details—likely the result of different artistic hands (Wu 1992a). The architectural setting of the central scene, for instance, is a courtyard complex foregrounded by a water landscape for both the north- and south-wall paintings. But the north-wall painting presents a more squarely constructed architectural space than the south-wall painting, because buildings in the former are fewer and less crowdedly positioned than in the latter, and fewer portions of the former's architectural backdrop are obscured by figures in the foreground. For the sake of concision and clarity, the following analysis focuses on the central scene of the north-wall painting.

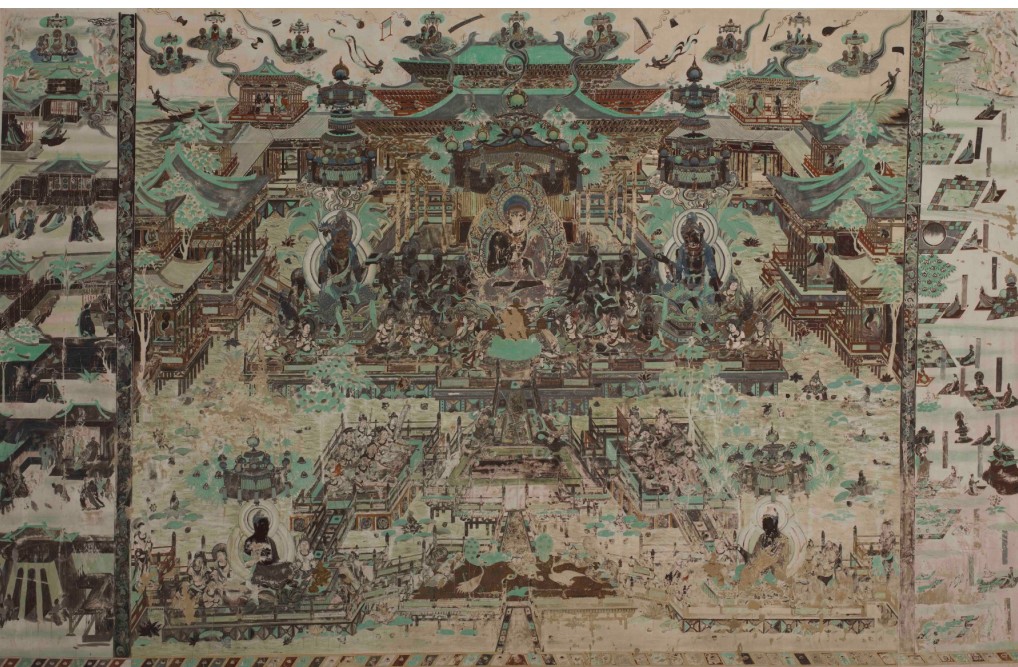

**Figure 2.** *Meditation Sūtra* transformation tableau. North wall of Mogao Cave 172. High Tang period. Mural painting. 400 (w) × 270 (h) cm. Photo Courtesy of Dunhuang Academy.

Amitābha Buddha (also known as Amitāyus), depicted as the central icon in the scene, is holding an assembly of buddhas, bodhisattvas, and heavenly musicians on railed



platforms that are raised above lotus ponds and surrounded by halls, pavilions, colonnades, and corner towers—a Chinese-style palatial complex that visualizes the medieval East Asian imagination of the Western Pure Land. Transformed buddhas, attending bodhisattvas, flying *apsaras*, and self-playing musical instruments hovering in the air, along with babies reborn via lotus flowers blooming in the ponds, are all in the process of joining the assembly. The infant-like aspirants are interlocutors for us, mortals of the actual world and beholders of the pictorial paradise.[5]

According to the scriptures, the way in which one enters the Pure Land is via a rebirth fueled by a persistent practice of contemplating the step-by-step manifestation of the Pure Land—the sixteen meditations.[6] It can take an extremely long time for one's lotus flower to blossom, depending on one's karmic debt. The *Meditation Sūtra* introduces a nine-leveled rebirth system called "the nine grades of rebirth" (*jiupin wangsheng* 九品往生) ([Wang 2003](), p. 693). While the rebirth system promises an all-inclusive salvation, only the lotus flowers for the aspirants of the upper four levels open immediately; aspirants of the other five levels must wait inside their lotuses for various durations. Those who belong to the lowest level of the lowest grade (*xiapin xiasheng* 下品下生), for example, must wait until twelve great *kalpas* have passed (*T* no. 365, vol. 12, p. 346, a 20–23).[7]

How could one endure the pain of not being able to reach the Pure Land while constantly contemplating one's being there? What roles did the Dunhuang cave art play in alleviating the pain of being so distanced? According to the *Meditation Sūtra*, a Pure Land practitioner may accomplish a spiritual journey by visualization, meaning "systematic building up of visual image, each as complete and precise as possible, in a sequence from the simple to the complex" ([Soper 1959](), p. 144).[8] Shandao 善導 (613–81), an eminent Tang monk and influential commentator on the *Meditation Sūtra*, emphasizes the importance of visualizing the holy beings and place with the "mind's eye" (*T*, no. 1959, vol. 47). Meanwhile, he suggests that visualization must be stimulated by concrete images. Furthermore, the making of Pure Land paintings promises "absolving one's multitudinous sins accumulated over eight billion *kalpas*" (除滅八十億劫生死之罪) (*T* no. 1959, vol. 47, p. 25, a09–10), and therefore accelerates the journey to Pure Land the lower-graded aspirants have to take. Image-based devotion has been an apparent motivation for constructing Pure Land caves and making Pure Land paintings in Dunhuang ([Wu 1992a](), p. 57).

For painters of the Mogao caves, the question of what image was made was no less important than why each image was made. Even if the relationship among the cave art, its visuality and ritual function, is viewed as complex and indirect ([Sharf 2013](), pp. 60–61; [Feng 2018](), pp. 66–75), ritual practices at least provide a lens to inspect the visual culture around the cult. Shandao's teachings and ritual texts used in medieval Dunhuang indicate that the Pure Land cult involved chanting the buddha' name, reading sūtras, performing eulogies, image worshipping, visual contemplation, and so forth (*T* no. 1753, vol. 37, p. 272, a, l.28-b, l.6, [Lin 2014](), p. 249). Some eulogies used in Pure Land rituals evoke a vivid image of "opening jeweled gates" and "right [at the moment] seeing" the buddha preaching ([Ren 1987](), pp. 573, 577).[9] This literary imagery presents a certain way of entering and seeing Amitābha's Pure Land, inviting us to consider the nonverbal invitation a visual image might present. Analogous to the eulogy, the painting would have presented an ideal image of the Pure Land in its makers' minds in a carefully designed way. The following analysis examines the way in which the visual representation of architecture attracts viewers into the presence of the pictorial paradise. Trace-copy line drawings, analytical drawings, and digital collage images are applied to explore the visual forms and visuality of the architectural images.

*2.3. An Architectural Approach: Visualizing the Pure Land in Its Entirety*

The painters of Tang China pulled out all the stops to make the Pure Land look real. They applied a proto-linear perspective to suggest a visual depth; they meticulously rendered the building structures to define the coordinate axes of a pictorial space; they carefully arranged the figures and atmospheric elements to indicate the foreground, the mid-

ground, and the fields of water and air. Most strikingly, by lining up three bridges and two terraces alternatively, the painters created a central path that guides one, or one's gaze, to meet with the Buddha (Figure 3). By means of a layered composition and a structured access, the image turns the temporal distance between the defiled world of ours and the Pure Land of Amitābha's into a spatial distance. In other words, the artists invented an *architectural* approach to the Pure Land.

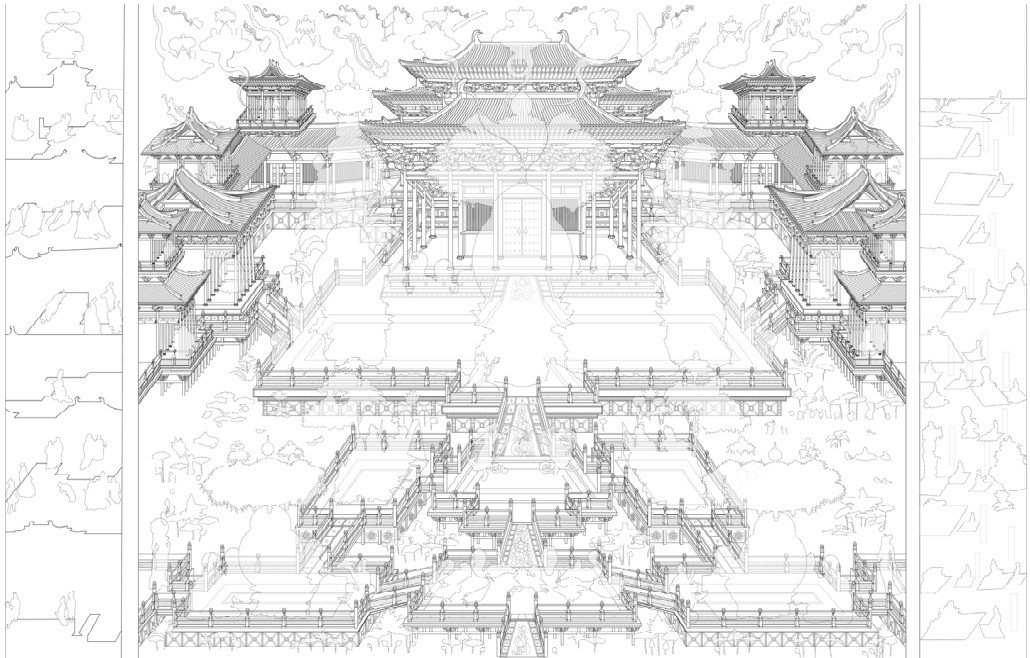

**Figure 3.** A trace-copy line drawing of the architectural setting in Figure 2. Drawing by Zhenru Zhou in AutoCAD.

This architectural approach carefully tailored the background to be analogous to the figure. An analysis of the pictorial composition will demonstrate the correspondence between the figural images and the architectural topography. Three circles of deities and aspirants are implied in the figures' composition (Figure 4); another three loops are revealed by the diagram of the architectural relationship (Figure 5).

First, the *shan* 山-shaped configuration of the Amitābha triad and two attending buddhas with entourages (nos. 1–5, connected by orange lines in Figure 4b) is echoed in the pictorial composition of the main halls, the corner pavilions, and the two sets of "a hall and two pavilions" (*yidian shuanglou* 一殿雙樓) on the sides (connected by yellow lines in Figure 5). Both mountain-shaped configurations pivot around the central icon and stabilize the core combination.

Second, the large circle of aspirants who encircle the buddhas and bodhisattvas (connected by green lines in Figure 4b) is parallel to the corridors that encircle the courtyard complex (marked by green lines in Figure 5). Both configurations form an outer circle of the multilayered complex.

Third, a small loop formed by the Buddhist figures and deities (marked by green lines in Figure 4b) is topologically identical with that which is composed of the terraces in the foreground (marked by green lines in Figure 5). The overlapped inner circles highlight a joyful assembly in the presence of the superior host, Amitābha Buddha.

The double configurations of triple circles visualize the multilayered topography of the Pure Land. According to the Pure Land scriptures, the Land of Bliss has "seven layers" (*qichong* 七重) of trees, jeweled nets, railings, and jewels, in addition to countless palatial halls (*T* no. 365, vol. 12, p. 342, b 02–09; *T* no. 366, vol. 12, 346, c 14–16).[10] In other words,

a visual template of Pure Land architecture has been invented to symbolize the hierarchy and emplacement of an ideal meeting with the Buddha.

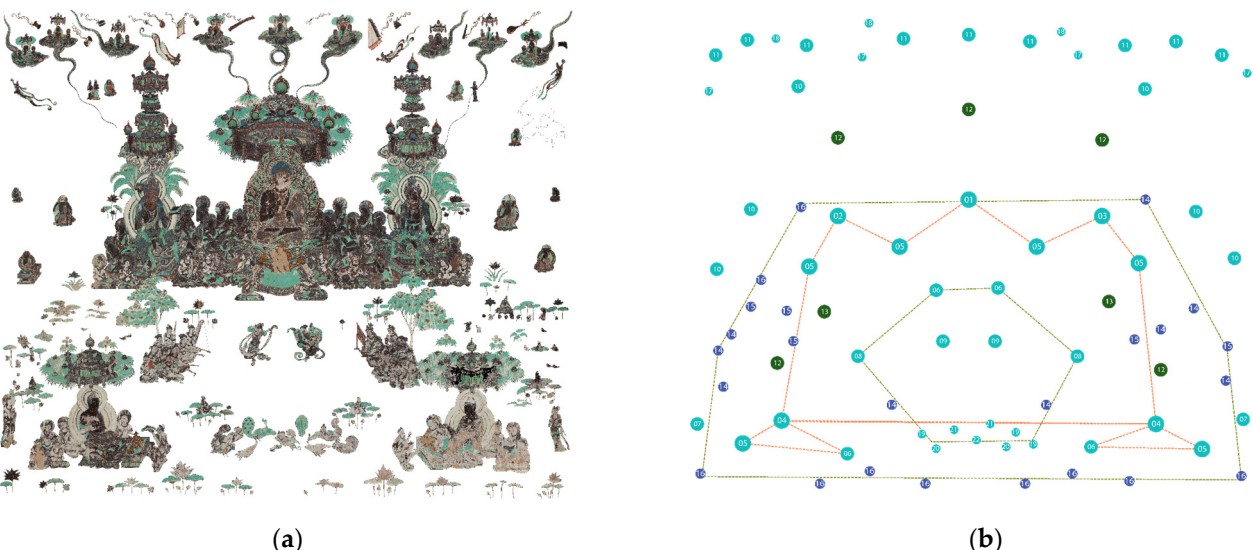

|(**a**)|(**b**)|

Legend (selected): (01) Amitāyus Buddha; (02) Avalokiteśvara Bodhisattva, (3) Mahāsthāmaprāpta Bodhisattva; (04) Additional Buddhas (05) Attending Bodhisattvas; (06) Offering Bodhisattvas; (08) Musicians; (09) Dancers; (11) Transformation Buddhas and Bodhisattvas (14), (15), and (16) Reborn aspirants of the upper, middle, and lower grades; (17) *Apsaras*; (19) *Kalaviṅka*; (20) Peacock.

**Figure 4.** The pictorial composition of figural images in Figure 2. (**a**) Isolated deity figures; (**b**) the locations and relationships between several figures. Digital photo collage and diagram by Zhenru Zhou in Adobe Photoshop.

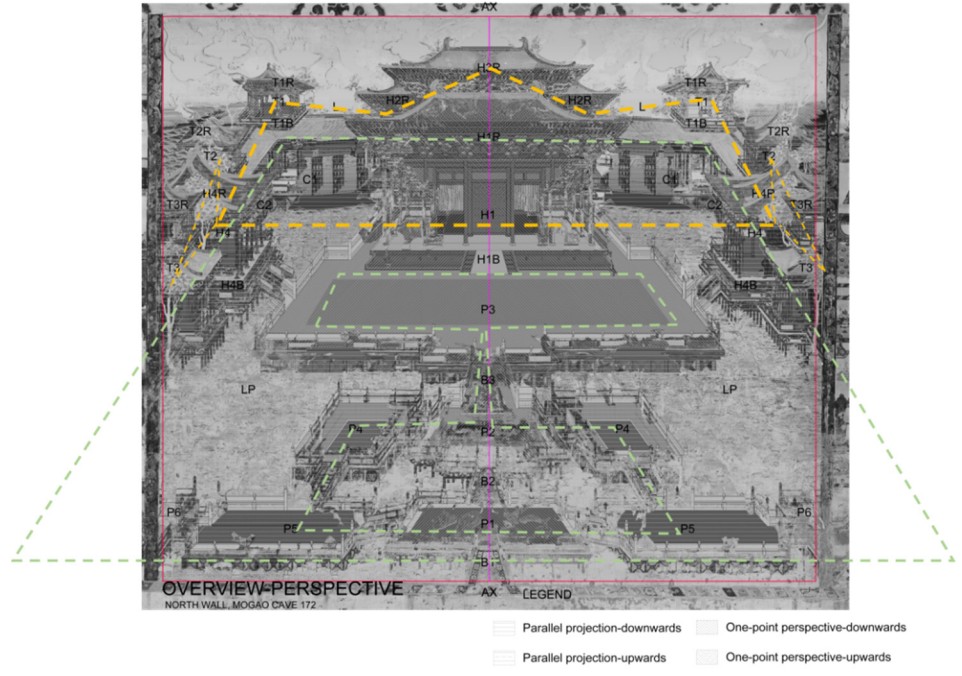

**Figure 5.** The pictorial composition of architectural images in Figure 2. Digital photo collage and diagram by Zhenru Zhou in Adobe Photoshop.

To better understand the historical visuality of this invention, it is necessary to provide an explanatory note about the use of perspective in premodern Chinese contexts. As often acknowledged, Chinese visual art enjoys a way of spatial representation distinctive from Western traditions (Zhang 2018, pp. 26–29). Therefore, here, perspective is broadly de-

fined as a method of representing three-dimensional forms and spaces on a pictorial plane. Formally speaking, the overall perspective of a Pure Land painting consists of two halves of isometric, oblique projection that mirror one another along a central vertical axis. At a local level, building forms are depicted from multiple viewing angles: roofs and façades seen from a frontal view (Figure 6a), eaves from below (Figure 6b), and grounds from above (Figure 6c). Because of the complexity of pictorial composition and architectural forms, no agreement has been reached about the terminology and visual logics of this hybrid manner of suggesting visual depths. The perspective is sometimes referred to as "the herring-bone perspective" because the vantage points are aligned along the central axis. Some scholars of Chinese paintings have also proposed calling it "parallel perspective" or "parallelogram perspective", while some others call it "perspective from point to point" because of the represented buildings (Zhang 2018, pp. 320–22; Zhao 2005, pp. 114–66; Fu 1998, pp. 75–94; Xiao 2019, pp. 321–47; Chung 2004, p. 27; Wang 2019; Wang and Li 2021). In any way, it is hard to ignore this Pure Land scene's strong allusion to a visionary space open to the viewer, an effect similar to modern "linear perspective" (Panofsky 1991, pp. 27–39; Gioseffi 1967).

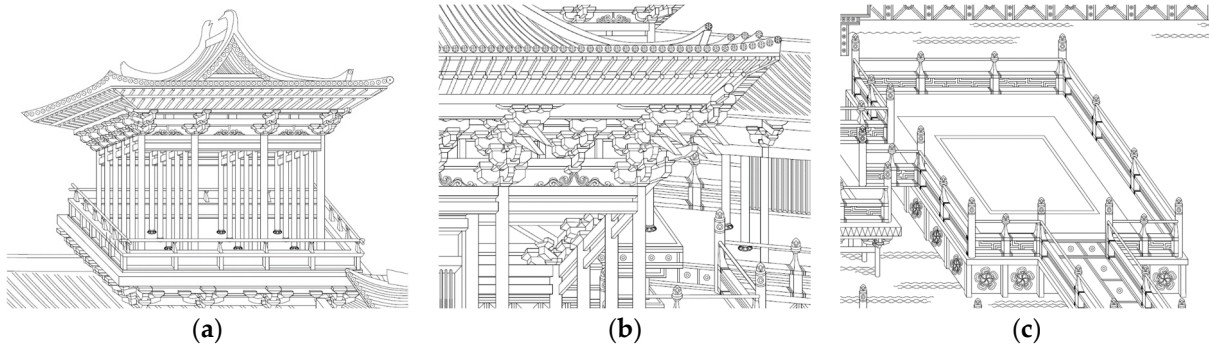

(**a**)          (**b**)          (**c**)

**Figure 6.** Details in Figure 3. (**a**) a corner pavilion represented in a frontal perspective; (**b**) the bracket sets and rafters of the main hall seen from below; (**c**) a terrace seen from above. Drawing by Zhenru Zhou in AutoCAD.

The conception of a painting as "a window on the world"—for which perspectival techniques were invented—pervades Western art history (Bird 2012), but it is less common in the premodern Chinese contexts. Many paintings, such as the two side panels of the Cave 172 tableau, are self-contained pictures that address the viewer as a witness rather than an active participant. According to art historian Wu Hung, the uncommon design of an iconic representation enforced by the proto-linear perspective suggests a direct relationship between the viewing subject and object. And the completion of the picture requires both that the Buddha exist within the pictorial space and that the viewer exist outside it (Wu 1992a, p. 54). Hence, the occasional suggestion of illusionist space would have been more visually striking for medieval Chinese viewers than for us, who are accustomed to post-Renaissance perspectival techniques.

An experiment of changing the architectural backdrop of the painting demonstrates specific effects of this proto-linear perspective. Assisted by digital imaging software, we replaced the original backdrop with a scientific, one-point perspective of the same building complex—the reconstruction design of which will be discussed later. The altered images display a more coherent spatial construct, but their pictorial compositions could no longer emplace the well-composed assembly. For instance, a bird's-eye view shows the overall layout at the expense of the canopy-like effect of the triply stacked roofs of the central halls (Figure 7). A one-point perspective at eye level conveys a sense of an architecturally encircled space but fails to include the assembly in the foreground (Figure 8). Because all parallel lines point to a single vanishing point, the adapted still images can comfort the eye but can never address all features in the mentally constructed environment of the

Pure Land. By comparison, the herringbone-perspective construct of the mural—meaning multiple vanishing points exist along the vertical central axis—allows almost all desired features of the Pure Land environment to be visualized (Figure 9).

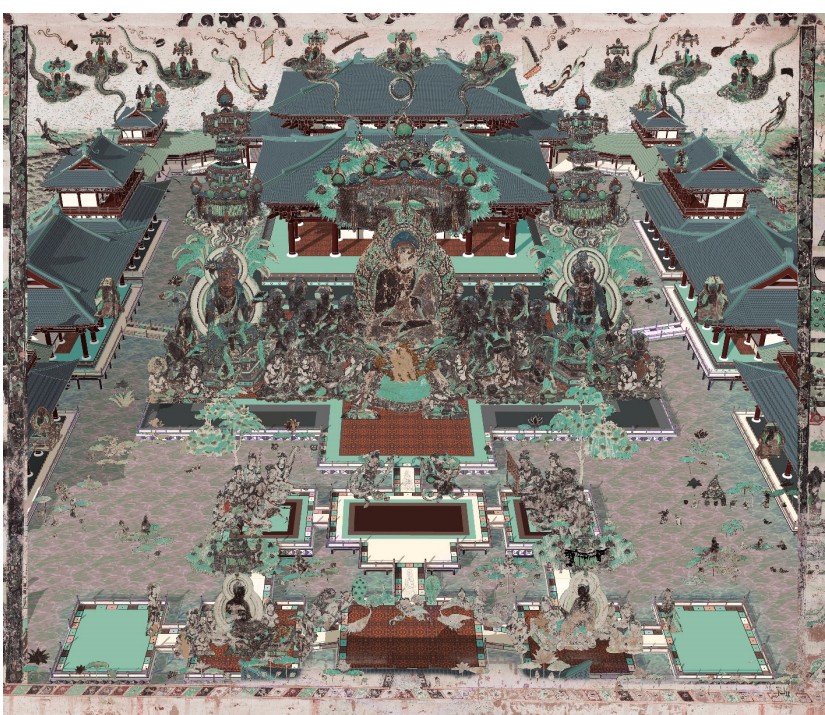

**Figure 7.** A bird's-eye view of the Western Pure Land: Zhenru Zhou's reconstruction design based on Figure 2. Digital photo collage by Zhenru Zhou in Sketchup and Adobe Photoshop software.

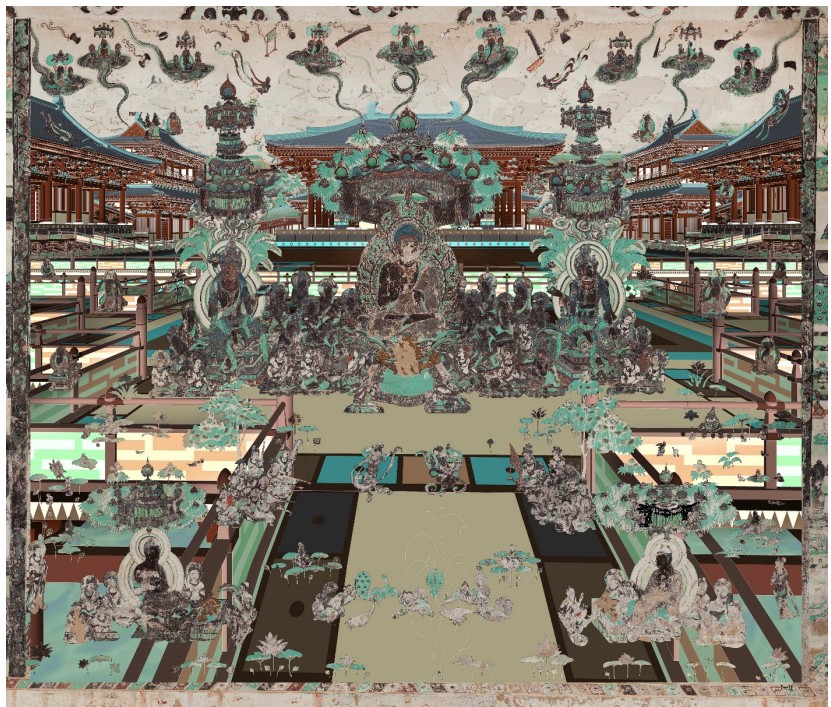

**Figure 8.** A one-point perspective of the Western Pure Land from a standpoint on the bridge looking toward the main terrace: Zhenru Zhou's reconstruction design based on Figure 2. Digital photo collage by Zhenru Zhou in Sketchup and Adobe Photoshop software.

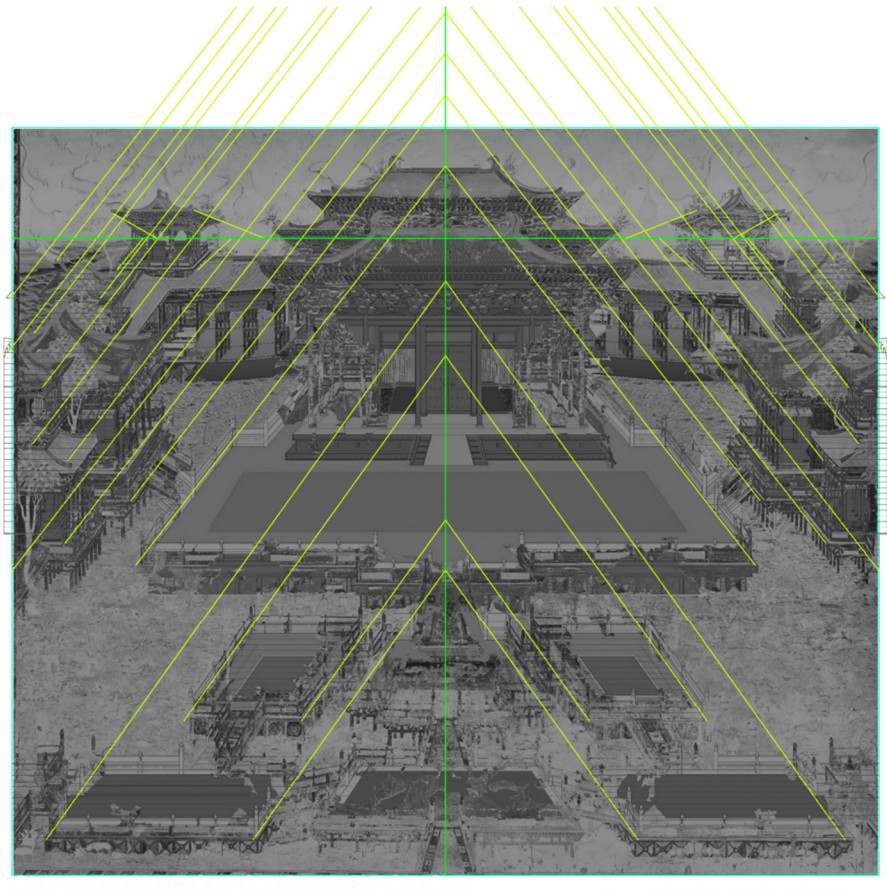

**Figure 9.** The herringbone-like construct of multiple "vanishing points" in Figure 2, with the architectural image completed. Diagram by Zhenru Zhou.

*2.4. Sequential Viewing: An Analogy of Sixteen Meditations*

The architectural approach was grounded on the notion of environment, which is crucial to the Pure Land meditation practice. Prior to visualizing the holy figures, one must evoke a vivid image of Pure Land topography. As the *Meditation Sūtra* prescribes, the sixteen topics for meditation are, successively, (1) the sun, (2) the water, (3) the ground, (4) the trees, (5) the pound of eight virtues, (6) the jeweled buildings, (7) the flower throne, (8) the image and (9) the body and light of Amitāyus Buddha, (10) Avalokiteśvara Bodhisattva, (11) Mahāsthāmaprāpta Bodhisattva, (12) comprehensive meditations related to rebirth assured, (13) miscellaneous meditations related to rebirth assured, (14) the upper levels of rebirth, (15) middle level of rebirth, and (16) lower levels of rebirth (*T* no. 365, vol. 12, p. 341, c l.29–p.346, a l.26).

Due to the sixteen topics' foremost importance for Pure Land petitioners, they were extensively discussed by Shandao in his commentary on the *Meditation Sūtra*, titled *Methods for the merit of samādhi by visualizing the sea-like Image of Amitāyus-Amitābha* (*Guannian Amitofo xianghai sanmei gongde famen* 觀念阿彌陀佛相海三昧功德法門) (*T* no. 1959, vol. 47). According to Shandao's commentary, the first seven topics are "dependent" rewards (*yibao* 依報) that help the meditator build up the Pure Land environment in the mind, the following six topics are "main" rewards (*zhengbao* 正報) that help the meditator envision the holy presence of the Amitābha triad, and the last three topics elaborate on the rebirth system (Feng 2018, pp. 257–60).[11] Simply put, visualization of the miraculous topography leads to contemplative confrontation with the holy presence.

These sixteen meditation topics were often illustrated in the *Meditation Sūtra transformation tableaux*—in this case, as the vertical panel painting accompanying the central

scene on the right-hand side (Figure 10). Following the top-to-bottom viewing sequence, a beholder's eye is naturally guided to the bottom register of the tableau, right next to the inviting architectural foreground in the central scene.

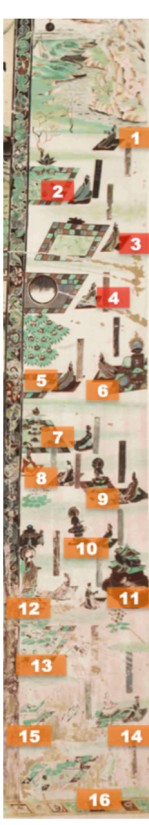

**Figure 10.** Sixteen meditations: *Meditation Sūtra* transformation tableau. North wall of Mogao Cave 172. High Tang period. Mural painting. Feng (2018, pp. 2–121, Figure 3–30).

The central scene, by means of its perspective construct, echoes the message the sixteen meditations panel conveys—that is, a sequential procedure of entering the Pure Land. Because the painting is too big to be grasped all at once by a worshipper in the cave, the worshipper is compelled to examine it part by part. Because multiple vanishing points exist along the vertical axis of the painting, the overall effect is similar to multiple-point perspective, which suggests the viewer's constant shift of position.[12] The viewing is accompanied by movements of an implied traveler whose steps the beholder travels to experience the visionary built environment. Based on the varied level heights of the vanishing points, we diagrammed the sixteen spatial units the imaginary traveler would traverse or see (Figure 11).

Looking at the painting at eye level or a bit downward (from a point 1–1.5 m above ground level), the worshipper naturally sees the lotus pond and terraces, from which the imaginary journey begins. To visualize the pictorial space, the worshipper first contemplates an imaginary traveler arriving at the Land of Bliss through the central-front bridge (scene 1). Then, the worshipper continuously contemplates the aspirant getting closer to the Buddha, passing through terraces and bridges one after another (scenes 2 through 5). This imaginary pilgrimage comes to a climax when the imaginary traveler arrives at the main terrace (scene 6).

As soon as the Amitābha triad manifests in front of the imaginary traveler, a transformation occurs in the worshipper's vision. Previously, the worshipper was looking down at someone else's movements, which was inferred from the high-view angles in scenes 1 through 6. Hereafter, the representation of the halls and pavilions is closer to the view at eye level, suggesting the worshipper's presence in front of the holy assembly. In addi-

tion to the direct relationship established between the viewer and the Buddha icon, the main hall behind the holy assembly looks just like what one might see when standing in front of a Buddhist temple (scenes 7 and 8). At that moment, the worshipper might even self-identify as the imaginary traveler in the pictorial space, because he or she sees what the latter would see. The worshipper has now become a witness of the Pure Land in the painting.

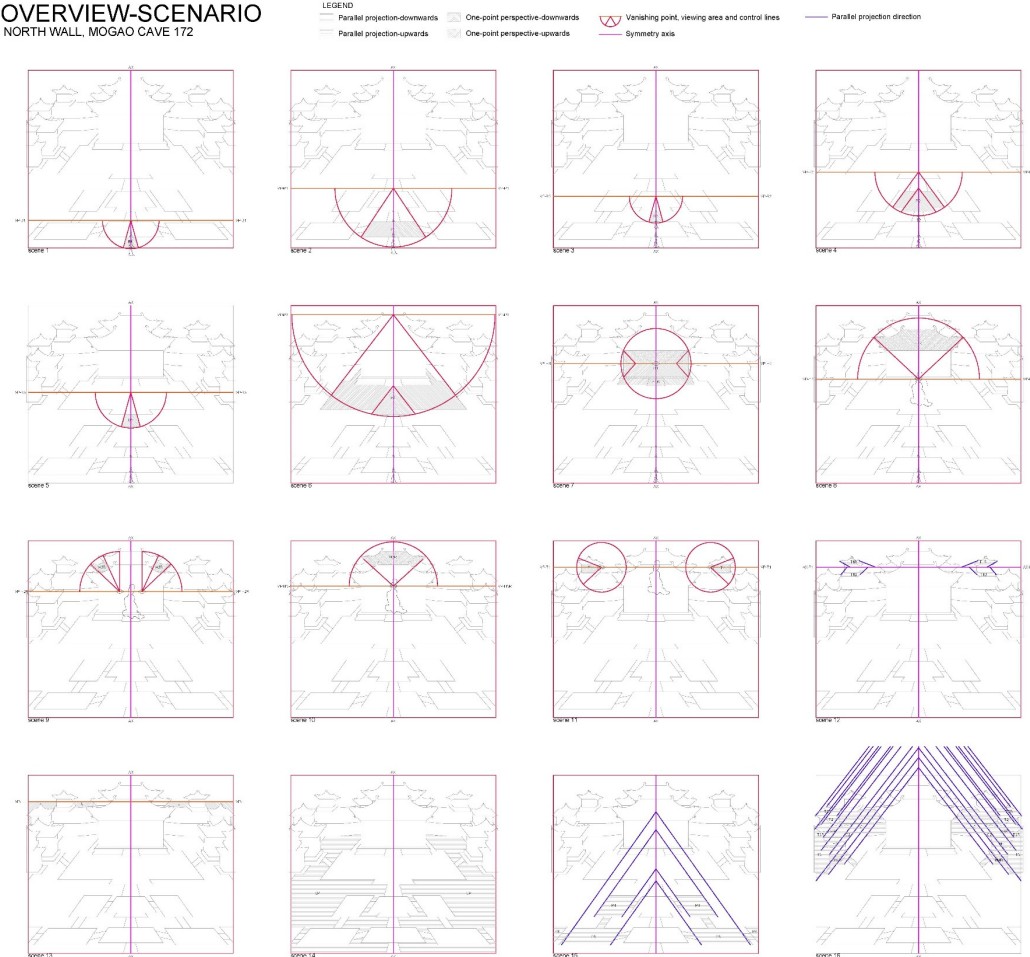

**Figure 11.** Sixteen steps of contemplating the pictorial space in Figure 2. Horizontal line is the level of the viewpoint of the imaginary traveler. Diagonal lines in a circle or semicircle are lines that are parallel in the pictorial space and represented as conjoining at the center of the circle (i.e., the "vanishing point"). Sets of diagonal lines mirrored along the vertical axis are in the direction of parallel projection. Diagram by Zhenru Zhou.

The subsequent scenes correspond to the worshipper's free roaming through the main hall and the hall(s) behind (scenes 9 and 10), gazing at the distanced land from the corner pavilions (scenes 11 through 13) or being suspended in midair and overlooking the entire assembly in the side halls and pavilions, on the terraces, and in the lotus ponds (scenes 14 through 16). In a word, the viewing sequence is analogous to an imaginary and transformative journey in the pictorial space. For the sake of more vividly conveying the aforementioned experience to modern eyes, we made a walk-though animation of a digital 3D model simulating the Pure Land environment, which can be found in the Supplementary Materials. Appendix A offers a close comparison between parts of the original painting and screenshots of the key moments in that walk-through animation.

Our theoretical reconstruction of the viewing procedure corresponds with the medieval Chinese monks' contemplation. A dreamy journey to the Pure Land is recorded in

*Biographies of eminent monks compiled during the Song period* (*Song gaoseng zhuan* 宋高僧傳) (*T* no. 2061, vol. 50; Li and Lü 1996, v2-2257; Theobald 2012). As the author, a tenth-century monk Zanning 贊寧 (919–1001), reports, two seventh-century monks, Qifang 啟芳 and Yuanguo 圓果, dreamed of visiting the Western Pure Land during a summer retreat at Wuzhen Monastery 悟真寺 in Lantian County. In their dream, Qifang and Yuanguo saw a great lotus pond and flew into a jeweled tent that was located on the east side of the pond. In the tent, they encountered monks who had been reborn there, Bodhisattva Avalokiteśvara, and Amitābha Buddha himself. After Amitābha assured them of a rebirth into his land, the jeweled tent carrying the holy beings departed toward the west. Then, moving westward, the two monks passed through three jeweled terraces that carried laymen, laymen and monks, and monks (*T* no. 2061, vol. 50, p. 863, b21–c14).

The anecdote reconfirms the Chinese imagination of the Pure Land environment, such as a ground as flat as mirror, a great lotus pond in which multiple terraces are erected, and jeweled canopies. More importantly, it illustrates the bodily movements of the imaginary visitors, who are spirits of the worshippers, through the terraces above in order to follow the trajectory of Amitābha. Should such a dreamy experience be pictured, the Pure Land transformation tableau offers a visual template.

### 2.5. Reflection on the Art Medium

Vivid as the mural painting is, any art medium has its limitations. While the pictorial "window" seduces a worshipper to look out into the space of a Buddhist paradise, the image-bearing surface—in this case, the wall of a rock-cut chamber—mercilessly denies any actual entrance to it. Some cave-makers in Tang Dunhuang were aware of the problem and sought to resolve it in the ritual programs of the cave to which the mural paintings belong. Hongbian 洪辯 (d. 862), an eminent monk of Dunhuang who patronized the construction of Cave 365 in 832–34, even took the chance of cave construction to give the following sermon to his disciples: "Clay niches are not substantial, but they may exert themselves to hold [the Buddha's teachings]. Bamboo and silk[-based artifacts] are not real, but they have the function of circulating [the teachings] (泥龕不實，而能作住持之功；竹素非真，而有流通之用)." (Zheng and Zheng 2019, 274–75)[13] The irreconcilable contradiction between the image and the art medium drove Hongbian to admit "clay niches are not substantial" and "Bamboo and silk[-based artifacts] are not real." Meanwhile, he and cave patrons alike did not lose faith in making caves, for that the cave provides a *real space*—to coin David Summers's (2003) term—in which a new dimension of visual art is made possible. The next section investigates the spatial mediums that Dunhuang cave-makers applied to resolve the problem of physical inaccessibility.

## 3. Spatial Imagery of the Pure Land Cave

The Pure Land image was not necessarily invented by the local artists and artisans of Dunhuang, but the painting medium is inseparable from the site—in this case, the cave temples cut into the living rocks. The architectural art of visualizing the Pure Land continued to evolve in situ at the Mogao caves. It reached an unprecedented degree of comprehensiveness by the end of the tenth century. The shared interest of residents and Buddhist societies of Dunhuang to synchronize the actual locale with the Buddhist paradise conspired to the spatial sequence of a cave and the grouping of multiple caves.

### 3.1. Spatial Components of Cave Suite 172/173

In analogy to the layered spaces of a pictorial paradise, the cave temple consists of a few architecturally defined spaces along the transversal axis (Figure 12). The spaces for a typical Tang cave like Cave 172, from outermost to innermost, are an antechamber, a corridor leading to the main chamber, a main chamber under a truncated pyramidal ceiling, and, cut on its rear (west) wall, a wide-open buddha niche (*changkou kan* 敞口龕). A series of cave spaces indicates a sequential viewing through which the mural paintings of the Pure Land are eventually reached.

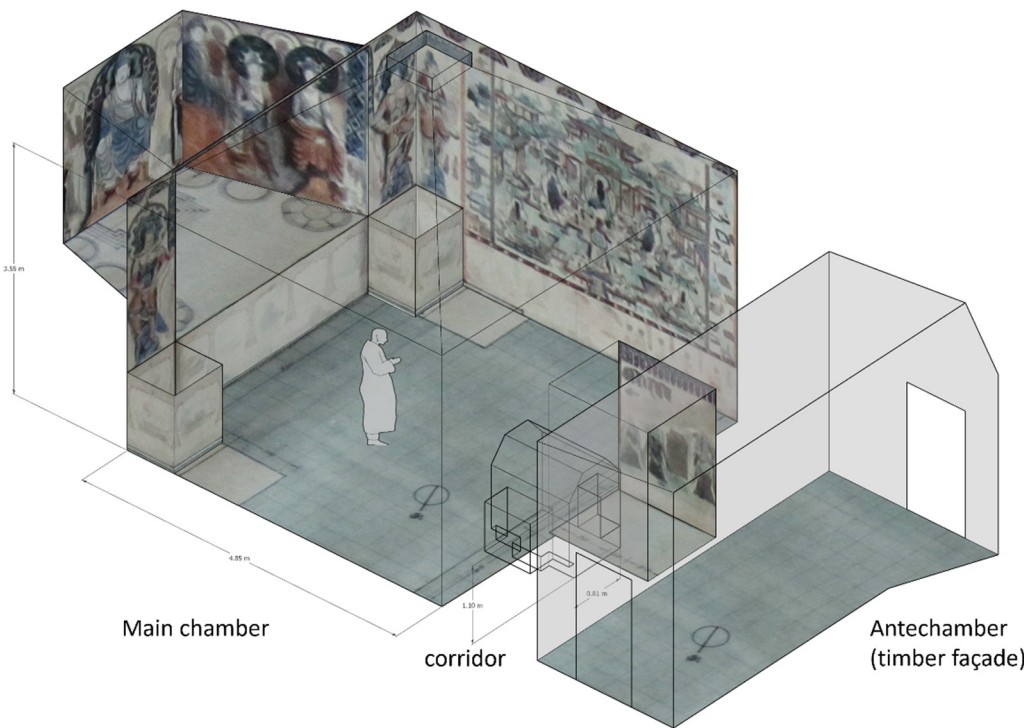

**Figure 12.** Isometric view of Mogao Cave 172 showing the dimensions of the main chamber wall and the ear-chamber wall. Drawing by Zhenru Zhou, texture after Sun Ruxian's rendering in Sun and Sun, *Shiku jianzhu juan*, 225.

Modifications in the subsequent periods added two significant features to this cave architecture. First, a miniature cave, numbered 173, was cut out on the south wall of the corridor of Cave 172 during a renovation in the late Tang period, circa. 851–900 (Dunhuang 1982, p. 69).[14] A miniature cave is not simply a small cave; it is a miniature version of a normal-sized cave. The overall size of its main chamber is no more than 0.9 (L.) × 1.2 m (W.) × 1.4 m (H.), but Cave 173 has nearly all the spatial components that Cave 172 has, including corridor, niche, ceiling, and buddha altar, all decorated with mural paintings (Figure 13). Because its corridor, sized 0.58 m (H.) × 0.3 m (W.), is not even large enough for a child to enter, the most likely possible way of construction would have been to cut out the miniature cave without a front wall, furnish and decorate it, and lastly fill the front wall while leaving a small opening. Once the corridor was refurbished, the dollhouse-like auxiliary cave became an integral part of this cave suite, enriching its spatial structure and diversifying its scales.

Second, the antechamber of Cave 172, along with those of its neighboring caves, was renovated around the tenth century (Dunhuang 1982, pp. 63–71). Two beam holes above the two upper corners of the corridor entrance indicate that Cave 172 used to have a three-bay-wide timber-framed façade (Gosudarstvennyĭ Ėrmitazh 1997–2005, v5, pl. 1). Fragments of murals were found on the cliff face above the row of caves that Cave 172 belongs to, suggesting pictorial decoration was part of their exterior appearance (Pan 1990, p. 64). Old timber structures in the district that Cave 172 belongs to have entirely perished, but remains a few hundred meters north shed light on the common design (Figure 14). Prior to entering the antechamber, a tenth-century worshipper would have confronted the cave's exterior glorified by a timber-framed façade and an open-air mural above the pitched roof.

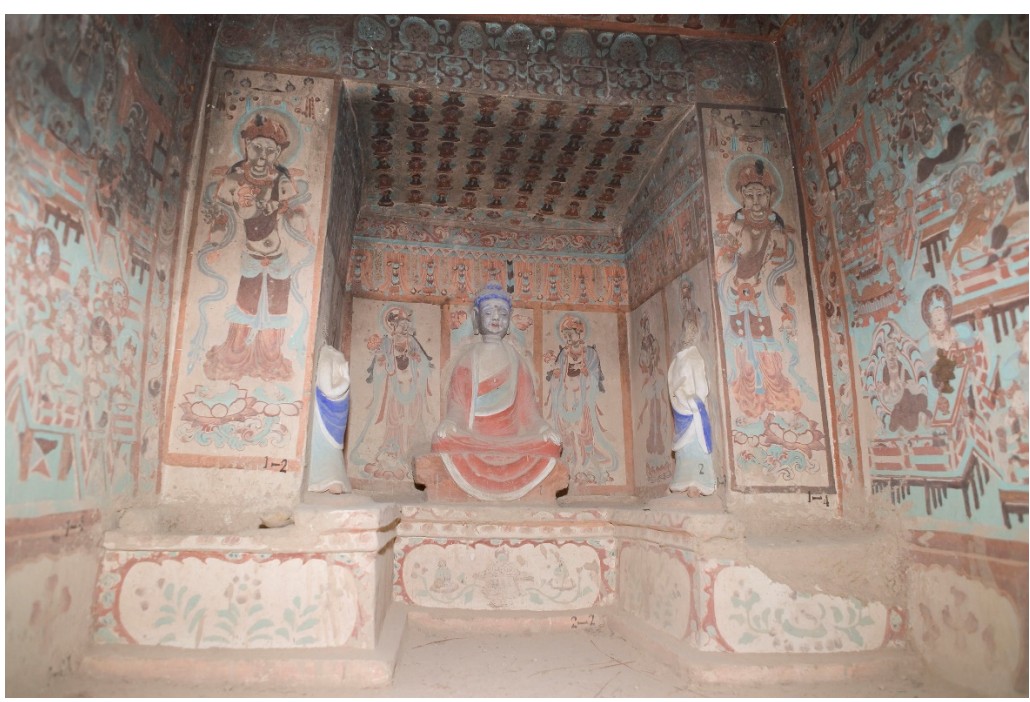

**Figure 13.** Cave 173, late Tang period, statues remade in the Qing period. Photo courtesy of the Dunhuang Academy.

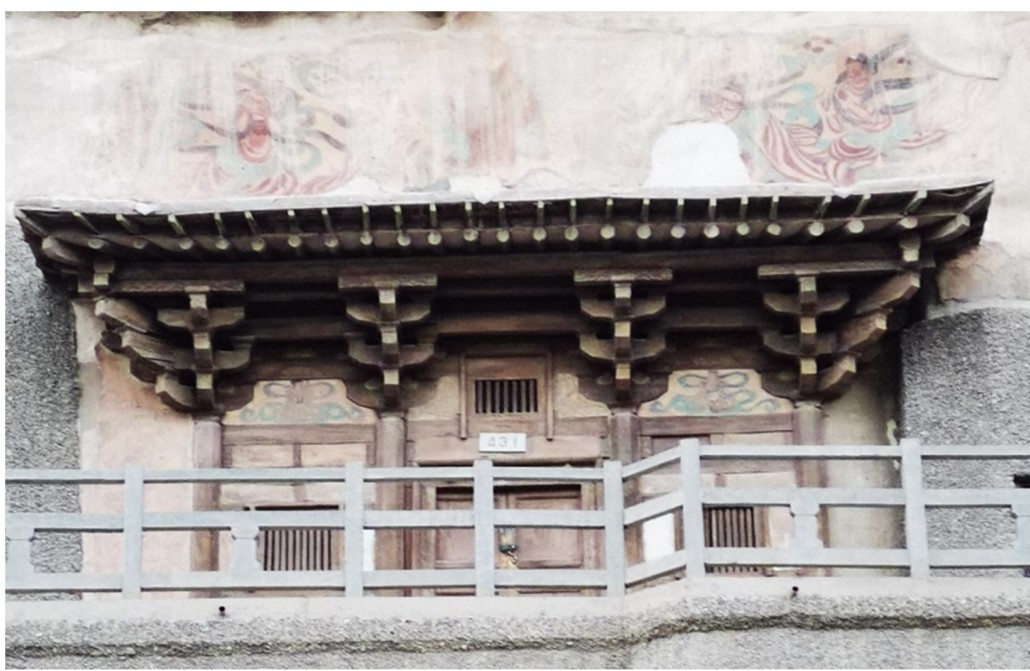

**Figure 14.** The timber-framed façade and exterior mural of Mogao Cave 431 showing three-step bracket sets, a three-bay façade, and an overhanging roof. Dated by inscription to 980 CE; 486 cm (w) × 142 cm (d) × 320 cm (h). Wood, mud brick, polychromic pigments. Photo by Zhenru Zhou, 20 January 2022.

Despite not being constructed or refurbished at the exact same time, the rock-cut chambers, timber-framed façade, interior and open-air murals, and polychromic clay statues constitute Cave Suite 172/173, and have defined what it looked like for the most part of its life. The composite materiality allows the cave to be an extraordinarily plastic medium; it not only allows one to enter but also conveys the image in a continuously flowing man-

ner (Wright [1930] 2008, pp. 72–73).[15] A visual analysis of Cave 172 with its auxiliary cave and the neighboring caves will reveal the plasticity of the cave medium.

### 3.2. Immersive Visual Space of the Main Cave

Above all, the main cave chamber, which is often smaller and more compact than the interior of a free-standing temple, intensifies a worshipper's confrontation with the Pure Land transformation tableaux on the north and south walls. The two paintings cover the entire width and extend from the top to about sixty centimeters above ground of the two opposite walls, which are 3.55 m tall, 4.85–4.95 m wide, and 5.1 m apart from each other (Figure 15) (Shi 1996, v2, Figure 247). For a worshipper who stands in the center of the main chamber, the painting takes up a field of view of about 87 degrees lateral by 59 degrees vertical.[16] This means that only a small portion of it can be grasped by the vision of central fixation (thirty degrees), whereas much is grasped by the peripheral vision (Spector 1990). It is the immersive visual environment that awakens the haptic longing for the Pure Land palaces.[17]

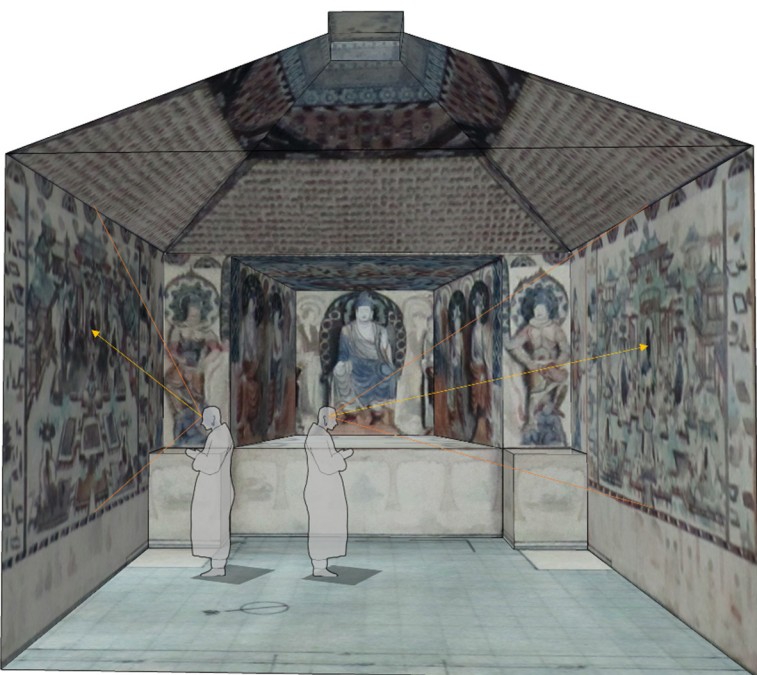

**Figure 15.** Potential viewing angles of the Pure Land paintings in Cave 172 (figure's height: 1.70 m). Drawing by Zhenru Zhou, texture after Sun Ruxian's water-color renderings in Sun and Sun, *Shiku jianzhu juan*, 225.

The buddha niche and the entrance corridor of a Pure Land cave temple are often part of the simulated palaces on water. In another High Tang cave, Cave 171, which is next to Cave 172 on the south side, an immersive environment for the worshipper to meet with Amitābha Buddha is enhanced by a "lotus pond and portal" simulated by the image niche, entrance, and ground pavement (Figure 16). As Feng acutely observes, the imageries of the Pure Land are not only painted on the three walls and sculpted in the niche, but also "colonize" the cave space in the case of Cave 171 (Feng 2018, pp. 222–27). This example reveals a mutual development of the pictorial space and the actual cave space.

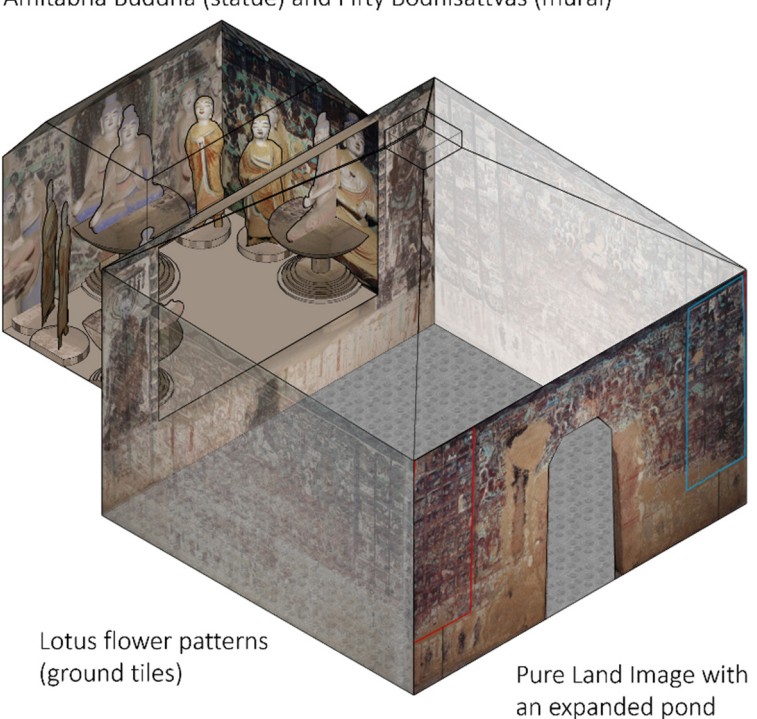

Amitabha Buddha (statue) and Fifty Bodhisattvas (mural)

Lotus flower patterns (ground tiles)

Pure Land Image with an expanded pond

**Figure 16.** Isometric view of the main chamber of Cave 171 showing the subject matter of the images along the niche-entrance dimension. Drawing by Zhenru Zhou.

### 3.3. Visual Tricks of the Auxiliary Cave

Furthermore, the function of the miniature Cave 173 parallels that of a side hall in the Pure Land building complex. The practice of excavating an auxiliary cave shrine onto the corridor or antechamber of a pre-existing cave temple was an effective way for the cave-makers to add a new "showcase" and to show respect for their forebears while maintaining the integrity of the cave temple (Ning 2004, pp. 65–75). This practice occurred at Mogao as early as the Northern Dynasties and became popular during the Guiyijun period. Zhenru Zhou's surveys of the Mogao caves have identified forty-three cave suites consisting of over a hundred caves (Figure 17, Appendix B).[18] They account for about one-fifth of the total number of image caves at Mogao, testifying to the common practice of building up a composite cave space.[19]

What is special in this case is that Cave 173 looks almost like a miniaturized replica of Cave 172; within a truncated pyramidal-ceiling cave no larger than 1.5 cubic meters, the rear wall is equipped with a buddha niche, and each of the side walls bears a Pure Land transformation tableau (Figure 13). The length, width, and height of the main chamber of Cave 173 are, respectively 18%, 24%, and 26% of those of Cave 172 (Shi 1996, v2, p. 193). In comparison to two other kinds of auxiliary caves—a niche enshrining small buddha images and a shadow cave enshrining a life-size monk statue—the miniature Pure Land cave displays a stronger manipulation of space construct. Through self-similarity and scaling, the cave suite encompasses two time-spaces that are concurrently independent from and resonate with each other. Although the patrons of Caves 172 and 173 have not been identified, the prolonged visual excitement is evident to a worshipper entering the cave suite; the visual encounter with the miniature cave prepares the worshipper to confront the main cave chamber along the central axis in a similar way in which the subsidiary halls in a palatial complex prepare a visitor before entering the main hall.

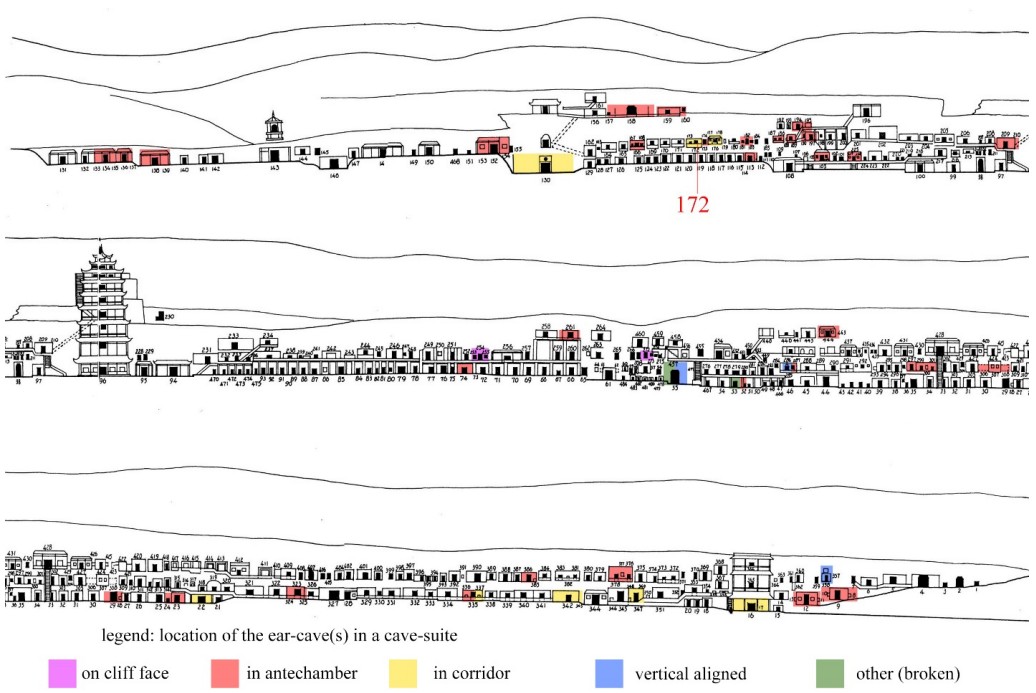

legend: location of the ear-cave(s) in a cave-suite

■ on cliff face    ■ in antechamber    ■ in corridor    ■ vertical aligned    ■ other (broken)

**Figure 17.** Distribution of cave suites at the Mogao Caves. Base map by Sun Ruxian. Annotation by Zhenru Zhou.

Before the construction of the auxiliary cave, the compositional principles of self-similarity and scaling were coded into the mural paintings inside the main chamber. All halls and pavilions represented in the Pure Land transformation tableau on the north wall share the same five-by-three-bay plan and have only two types of roofs: hipped, and hipped and gabled. It is through multiplication, scaling, rotation, and stacking the basic units that the palatial complex is composed (Figure 18). If we take the width of the main hall, which is depicted right behind Amitābha Buddha, to be 1, then the rear hall, the side halls, the corner pavilions, and the pavilions flanking the side halls are, respectively scaled by 73%, 60%, 29%, and 28%. The flanking and corner pavilions are about one-quarter of the size of the main hall. Likewise, the auxiliary cave is about one-quarter, by width and by height, of the main chamber. This is perhaps not just a coincidence. As Luke Li has discussed elsewhere, miniaturization has been an effective way of creating spatial layers in Chinese religious architecture. A Buddha Hall with its furnishing may utilize three scales—the building scale, 1/2–1/4 of it, and 1/10 of it—to assist the visual imagination of heavenly palaces (Li 2020, p. 28).

A small cave is not as simple as it appears. Although unenterable, it echoes the visual programs of the main cave and constitutes a cave suite that aligns with the compositional principles of the Pure Land painting. As Zhang Yingrun 張盈潤 (ca. 927–50), a Dunhuang layman who visited the Mogao caves in 939, recalls, "Doubly opening the rock chambers, I worshiped the thousand honored ones as if in the immortals' realm [*chongkai shishi, li qianzun si dao Penglai* 重開石室, 禮千尊似到蓬萊]" (Dunhuang 1986, pp. 53–54).[20] The composite cave space which could be "doubly open[ed]" alludes to the multilayered environment of the Pure Land.

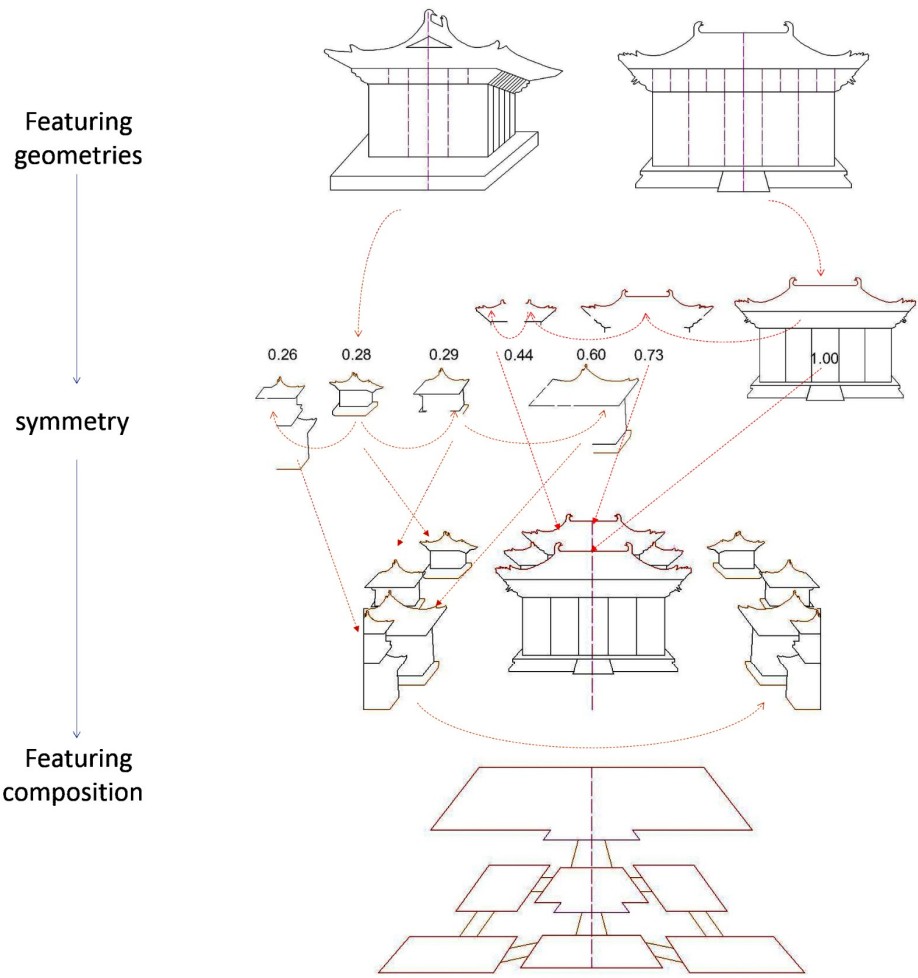

**Figure 18.** The building prototype and the methods of scaling and positioning for generating the architectural complex in Figure 2. Diagram by Zhenru Zhou.

### 3.4. Open-Air Murals

Lastly, the open-air murals and the timber-framed façades allow the image to emerge from the cliff surface. Remains of the mural on the cliff above Cave 170, a cave adjacent to Caves 171 and 172 on the south side, depict a hipped roof with flaming jewels above its side pitch and a cinnabar-colored orb encircled by a green and red ring (Figure 19). Judging from the two rectangular holes on the antechamber wall, which were made to hold beams, this pictorial roof painted around the tenth century formed a backdrop for an actual roof that was placed atop the timber beams. In other words, the composition would have closely resembled the double or triple layers of roofs above the main Buddha icon in Pure Land paintings commonly seen after the eighth century. The image of overlaid roofs may represent a set of halls arranged either one in front of another (Figure 2) or one above another (Figure 20). Unlike a mural painting or a silk painting, the composite image that emerges from the combined media of cliff murals and timber architecture is physically accessible. Just as the miniature cave on the corridor of Cave 172 refers to the main chamber, the double-roofed façade of Cave 170 is a prelude to the pictorial palace one expects to see inside the chamber.

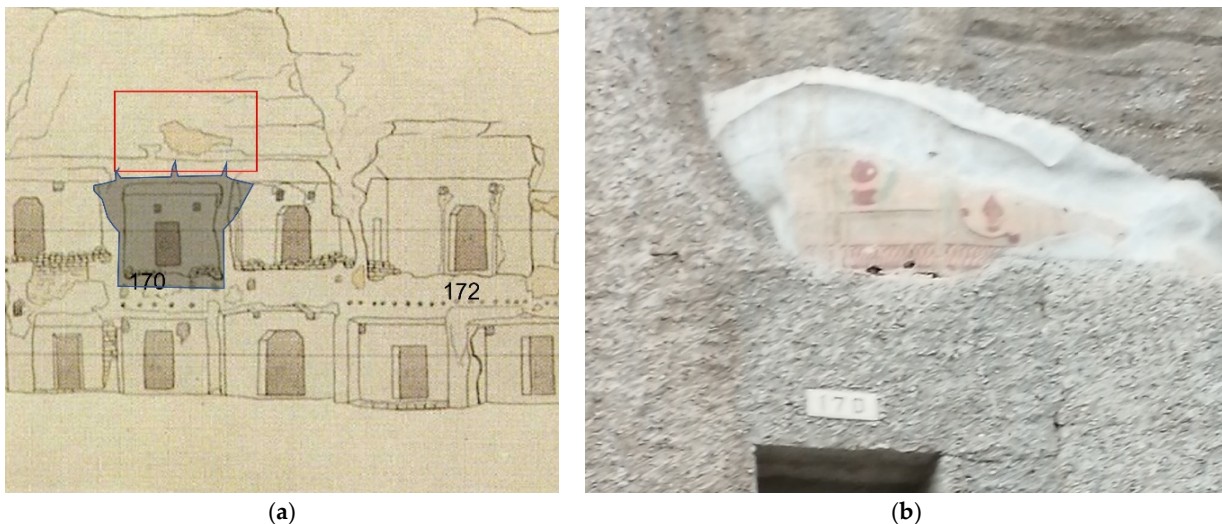

|     |     |
| :-: | :-: |
| (**a**) | (**b**) |

**Figure 19.** Open-air mural above Cave 170. (**a**) Location of the mural (in red rectangular frame next to gray shade of a hypothetical façade added by Zhenru Zhou) in Oldenburg's 1914–15 rendering (Gosudarstvennyĭ Èrmitazh 1997–2005, v5, pl. 1); (**b**) a recent photograph. Photo by Zhenru Zhou, August 2019.

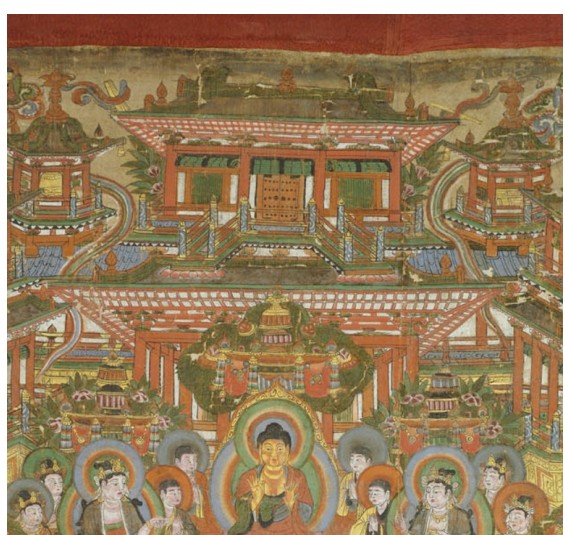

**Figure 20.** A detail of a *Meditation Sūtra* painting showing a two-level pavilion as the backdrop of a Buddha preaching scene. Silk painting, ninth or tenth century. Discovered in Mogao Cave 17. In the collection of the Guimet Museum (MG 17673). Digitized and made available by the International Dunhuang Project, https://idp.bl.uk/collection/F67D8E2A42102A44A2CC0D37DD4C6271/?return=/collection/?term=17673 (accessed on 26 February 2024).

Through visual alignment, liminal location, and nearly life-size images, the open-air mural mediates between the natural cliff and the built environment and between the pictorial and actual topographies. The cave complex was connected by timber-structured porches and pavilions by the end of the tenth century. In addition, a consensus is that the megastructure was decorated by a long stripe of open-air murals (Pan 1990; Ma 1996, p. 113).[21] Although only fragmentary traces of the stripe are preserved near Caves 170 to 173, a longer section of the stripe, which is located about twenty meters north from them, gives us a sense of the close relationship between the exterior mural and the façades (Figure 21).[22] The remaining murals were painted on a horizontal cliff area about 1.5 m tall right above the nonextant overhanging roofs of the second-level Caves 181 to 185, on which the beam holes are visible. The lateral connection of caves on the same level was visually

augmented by the open-air mural stripe. Such a spectacular scene must have evoked a wondrous feeling in the aforementioned Zhang Yingrun's mind. This feeling is recorded in his inscription outside the ante-hall of a cave below: "Connected with the passageways of pavilions on both sides, I visited the ten thousand images as if in the Buddhalands [*pangtong gedao, xun wanxiang rutong foguo* 傍通閣道, 巡萬像如同佛國]" (Dunhuang 1986, pp. 53–54).

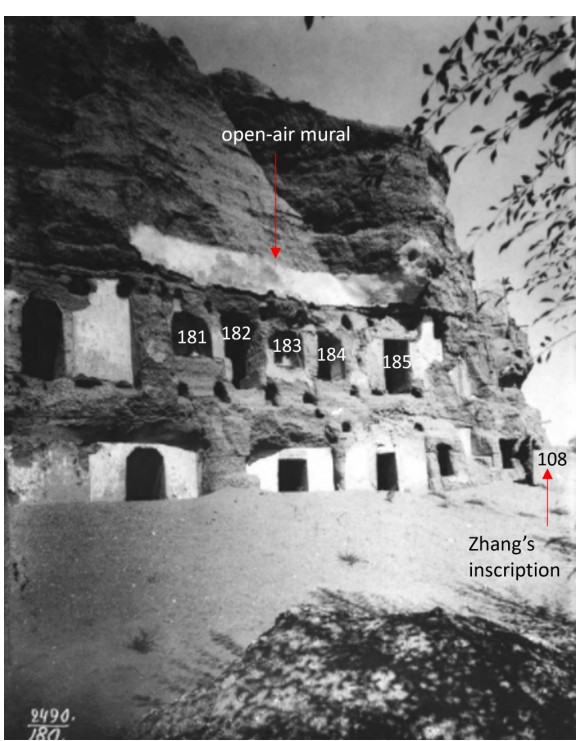

**Figure 21.** A stripe of open-air mural remaining above the antechambers of Mogao Caves 181–85. Photo by Oldenburg expedition team, 1914–15. After Gosudarstvennyĭ Ėrmitazh, *Eluosi guo li Ai'ermitashi bo wu guan cang Dunhuang yi shu pin*, 3:13.

## 4. Summary

This article has examined the historical visuality of the architectural imagery in Mogao Cave 172 and its extension in real spaces. First of all, the Tang-period visual paradigm tended to represent palatial complexes in the Pure Lands, inventing an architectural approach to bridge the distance between them and the world in which humans live. Well-composed pictures, such as the *Meditation Sūtra* transformation tableau in Cave 172, made this architectural stage look real while revealing all desirable features and a multilayered composition. The mural painting, conditioned by the picture size and cave space, was more likely to be viewed in a sequence analogous to the sixteen meditations. The spatial construct of the image made the viewing procedure transformative.

Furthermore, the article explored how the compositional principles of the Pure Land paintings could have assisted the transformation of the cave spaces and cliff site. It concerned not just the rapid development of architectural backgrounds in Pure Land scenes around the eighth century, but also the continued localization and actualization of the images at the Mogao site. This thorough transformation is inseparable from cave-making practices in the subsequent centuries, namely, the spatial intricacy of cave architecture introduced by cave suites since the ninth century, and the systematic refurbishments of the cliff face that turned it into a canvas of visionary topography in the tenth century. The iconography of the open-air murals is simpler and more generic than their counterparts inside the caves, and auxiliary caves provide multiplicity to the spatial structure. They, nonetheless, continued the theme of the pictorial image and enriched its expressive forms.

The modified Mogao Cave 172 continues to testify to the suggestive power of Pure Land art and gives the best clue to the sequential formulation of the paradisiacal image in pictorial, plastic, and architectural mediums.

**Supplementary Materials:** The following supporting information can be downloaded at: https://www.mdpi.com/article/10.3390/rel15030329/s1, Video S1: Walk-through animation of a Pure Land courtyard based on a painting in Mogao Cave 172.

**Author Contributions:** Conceptualization, Z.Z. and L.L.; methodology, Z.Z.; software, Z.Z.; validation, Z.Z.; formal analysis, Z.Z.; investigation, Z.Z.; resources, Z.Z.; data curation, Z.Z.; writing—original draft preparation, Z.Z.; writing—review and editing, Z.Z.; visualization, Z.Z.; supervision, L.L.; project administration, Z.Z.; funding acquisition, L.L. All authors have read and agreed to the published version of the manuscript.

**Funding:** This research was funded by "Chronological Study on Historical Resources of Architecture Image in Song Dynasty (兩宋圖像史料所見建築史料編年研究)", Subtask in "Chronological Study on Historical Resources of Architecture in Song Dynasty (兩宋建築史料編年研究)", Major Project, the National Social Science Fund of China (國家社會科學基金重大專案), grant number: 19ZDA199, and "A Research on the Decoration and Color of the Timber-framed Buildings during 7 ~ 13th Centuries, Based on the Ancient Wooden Relics, Drawings, and Brick Buildings in Imitation of Wooden Structures in *Yingzao Fashi* (基於《營造法式》的唐宋時期木構建築、圖像及仿木構建築中的建築裝飾與色彩案例研究)", General Project, the National Natural Science Foundation of China (國家自然科學基金面上專案), grant number: 51678325. The APC was funded by the School of Architecture, Tsinghua University.

**Institutional Review Board Statement:** Not applicable.

**Informed Consent Statement:** Not applicable.

**Data Availability Statement:** No new data were created.

**Acknowledgments:** This project was initiated during Zhenru Zhou's M. Arch studies at School of Architecture, Princeton University in 2016, and further developed into a section in her dissertation at the Art History Department, the University of Chicago in 2023, before it takes the current shape. Its development has benefited from comments by scholars at those two institutes, including Jesse Reiser, Elizabeth Diller, Jerome Silbergeld, Dora C.Y. Ching, Wu Hung, Wei-Cheng Lin, Katherine R. Tsiang, Anne N. Feng, Zsofia Valyi-Nagy, and Maggie Borowitz.

**Conflicts of Interest:** The authors declare no conflicts of interest. The funders had no role in the design of the study; in the collection, analyses, or interpretation of data; in the writing of the manuscript; or in the decision to publish the results.

## Appendix A. The Imaginary Journey in the Pure Land Transformation Tableau in Mogao Cave 172

The sequential viewing of the Pure Land painting is paralleled with a walk-through experience of the Pure Land architecture it represents. Since the herringbone perspective has multiple vanishing points, it suggests a viewing experience by moving through the space. For example, the vertical axis of the picture indicates that the main route is along the central axis in the pictorial space. And the shifting perspectives toward those buildings and architectural components are designated according to the perspective analysis of the painting. By adopting contemporary techniques of visualizing architectural spaces (with software such as Sketchup, AutoCAD, and Adobe Photoshop), I made a walk-through animation (Supplementary Materials: Video S1) to represent the bodily experience of the Pure Land topography in a way familiar to a present-day audience.

The diagrams paired with scenes from a walk-through animation show how the viewing experience of the painting evokes an imagination of a bodily experience into the Pure Land topography (Figure A1): A beholder arrives at the Land of Bliss through the central-front bridge (scene 1). Then, he or she goes closer to the main shrine, passing through terraces and bridges one after another (scenes 2 through 5). This imaginary pilgrimage comes to a climax when he or she stands on the main terrace (scene 6). The viewer then

looks up at the main hall and enters it (scenes 7 and 8), then passes through the main shine and similarly looks up at the hall in the rear center (scenes 9 and 10). The viewer may climb up to the corner pavilions and gaze at the distant landscape (scenes 11 through 13). He or she can also stay suspended in the air with the celestial beings attending the meeting and look from above at the subsidiary halls, the terraces, and the lotus ponds (scenes 14 through 16).

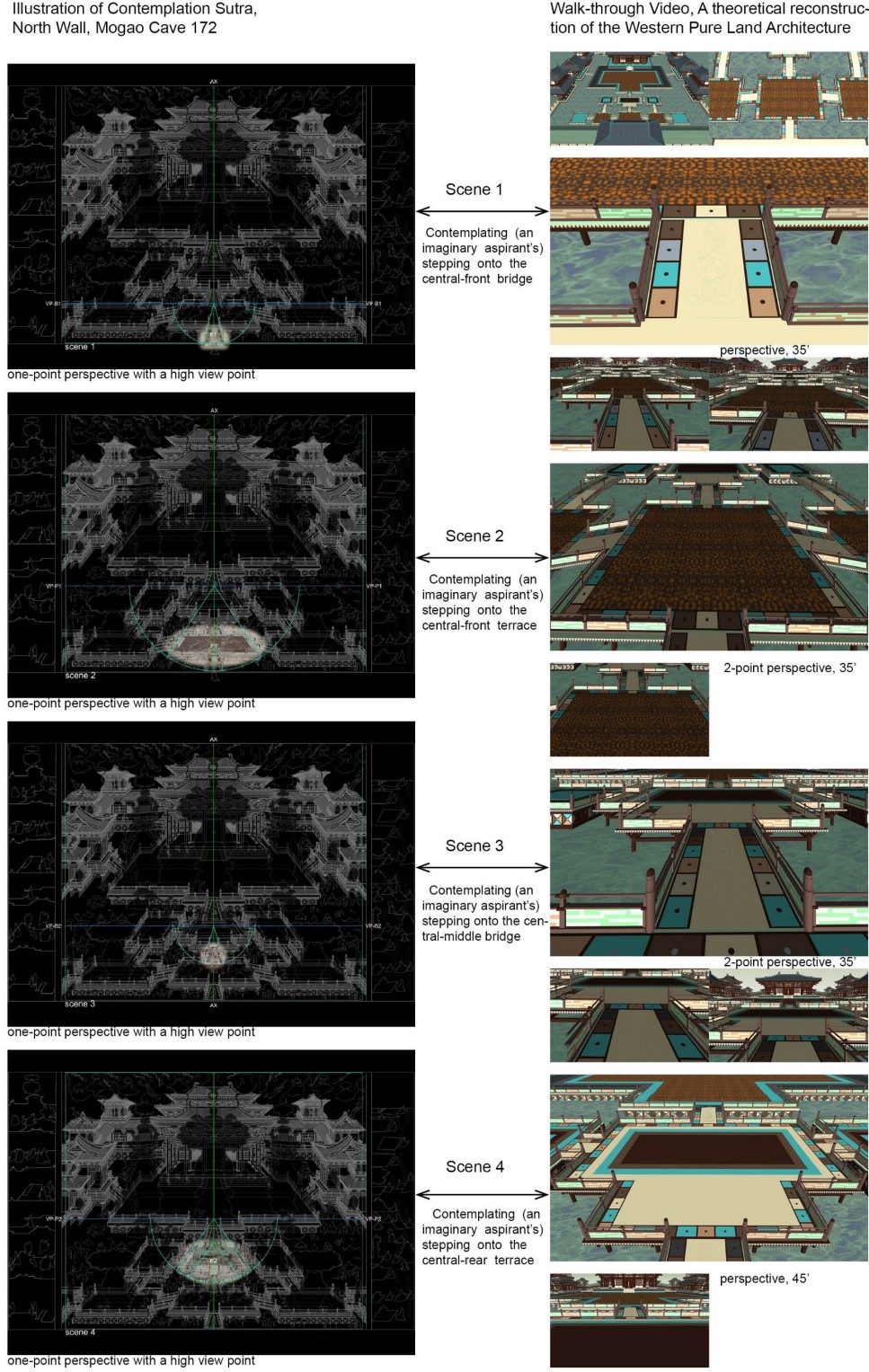

**Figure A1.** *Cont.*

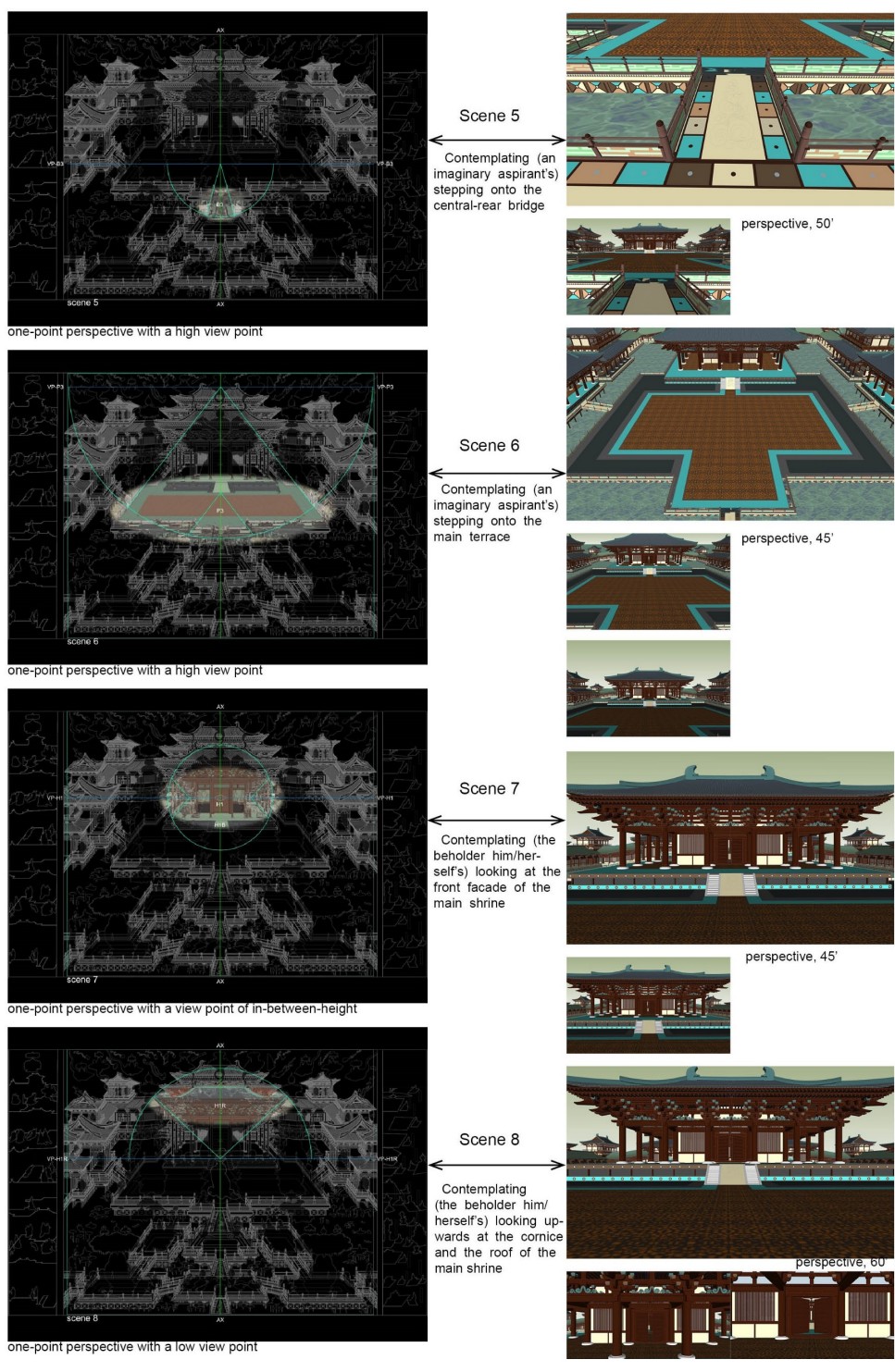

**Figure A1.** *Cont.*

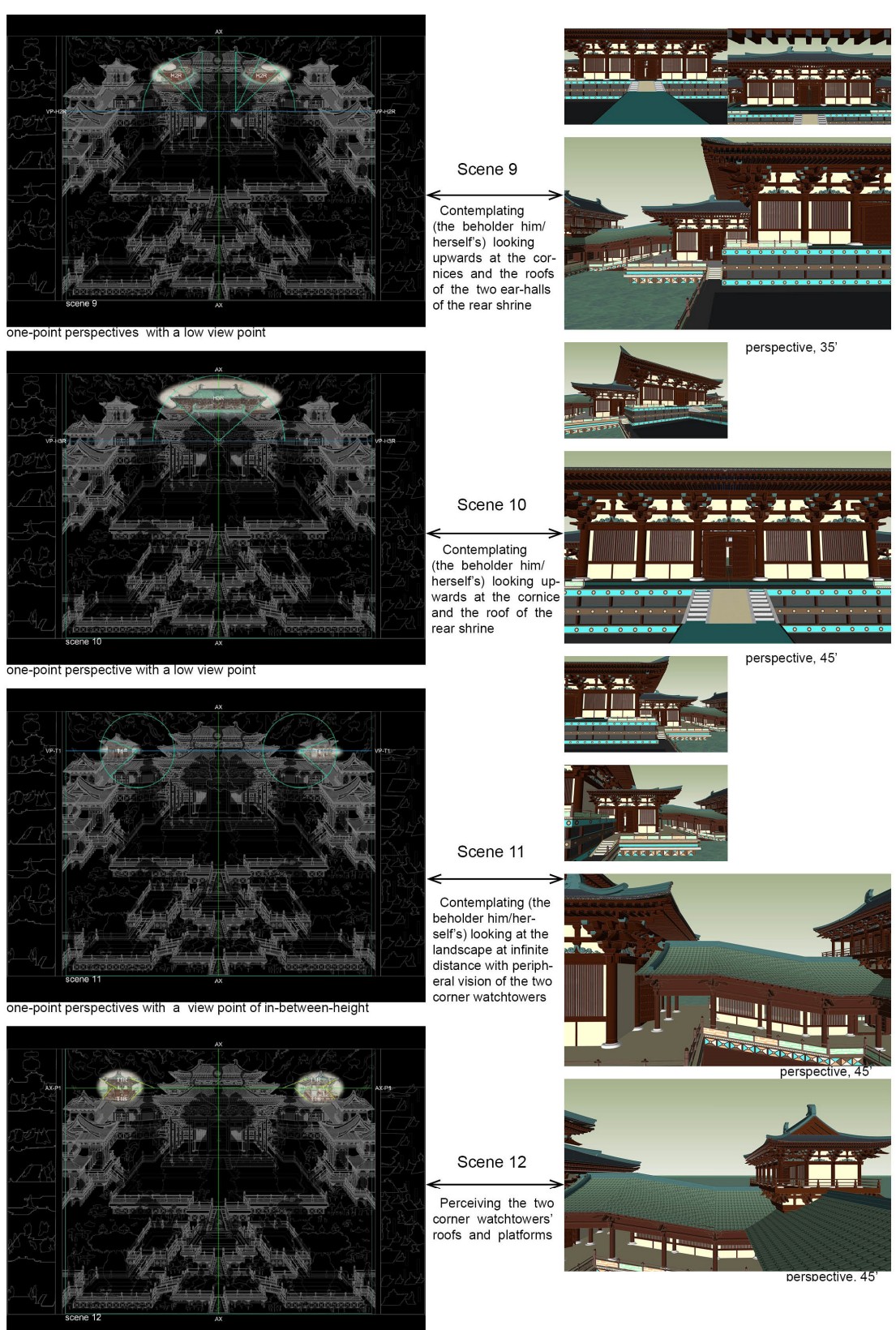

**Figure A1.** *Cont.*

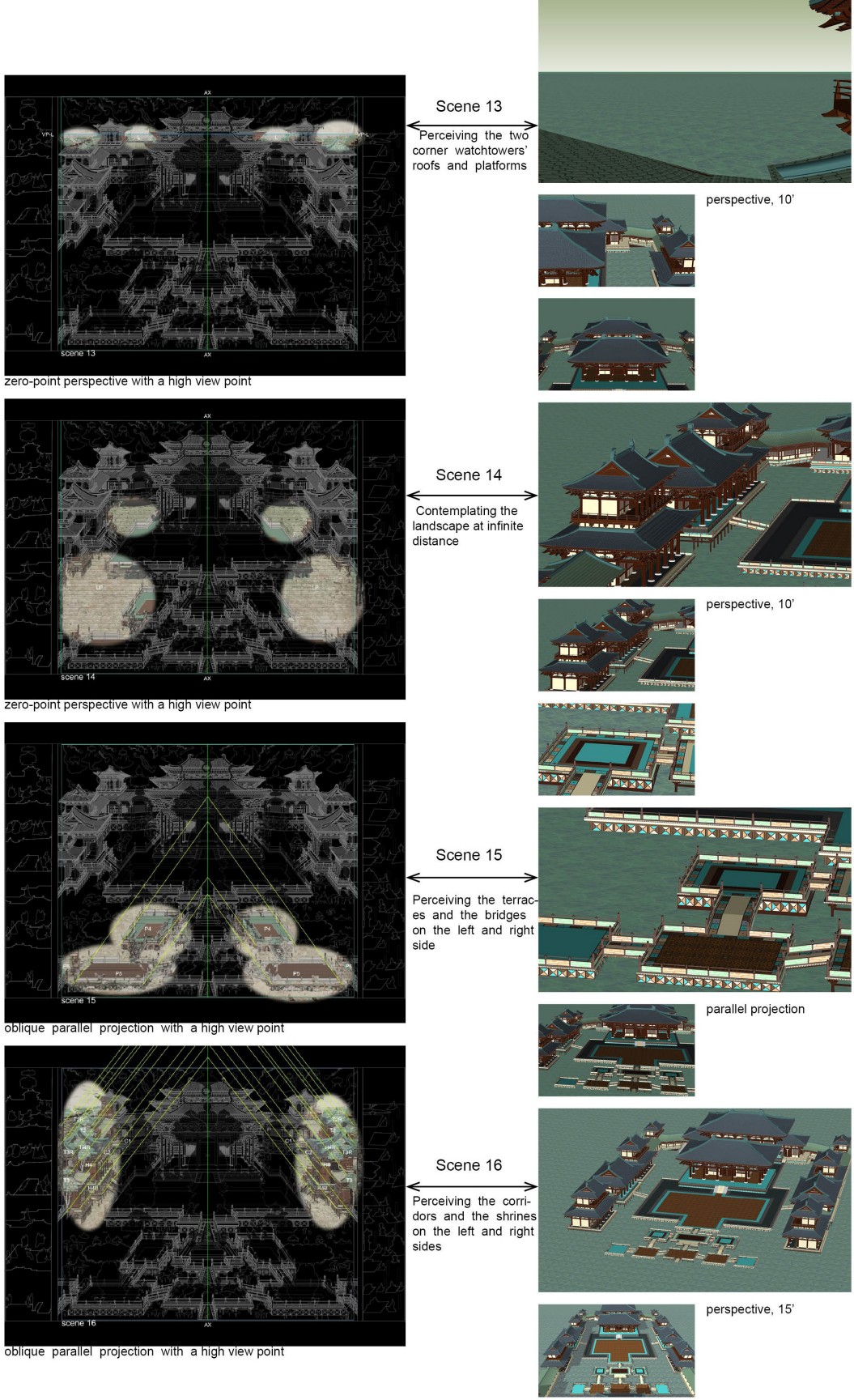

**Figure A1.** The sequential viewing of the Pure Land painting (**left**) paired with scenes from a walk-through experience (**right**).

## Appendix B. Cave Groups at the Mogao Caves

**Table A1.** Cave numbers and plan drawings of cave suites at Mogao.

| Set No. [1] | Cave Nos. [Main Cave(s)/Auxiliary Cave(s)] | Location of the Ear Caves or Auxiliary Caves [2] | Construction Periods [Main Cave(s)+(Auxiliary Cave(s)], Renovation Periods of the Main Cave | Brief Description of the Formation of the Cave Suite and Special Function of the Ear Cave If Applicable |
|---|---|---|---|---|
| 1 | 272/(273+272a) | Cliff face | Northern Liang+(Northern Wei) | The two niches enshrining statues of meditating monks were added later. |
| 2 | 254/(253+255) | Cliff face | Northern Wei+(Sui), Sui renovation | The two ear caves were added during the renovation of the corridor to the main cave. |
| 3 | 285/(286+287) | Antechamber, west wall above and north | Western Wei+(Western Wei+Early Tang), mid-Tang, Song, Xixia, Yuan renovations | Cave 286 was adapted from a high window above the corridor to Cave 285 during the construction of the latter, whereas Cave 287 was added later. |
| 4 | 307/(306+308) | Antechamber, south and north walls | Sui+(Sui), Five Dynasties and Xixia renovations | The three caves were made and renovated in the same periods. The niche of the main cave was added later. |
| 5 | 297+299+301/300 | Antechamber, west wall middle | Northern Zhou+(late Tang) | Cave 300 was added between Caves 299 and 301, which were adapted to share an antechamber. |
| 6 | 209/(210+209a) | Antechamber, south and north walls | Early Tang+(early Tang), Five Dynasties renovation | The main cave and at least one of the ear caves were made at the same time. I attributed number 209a to a half-damaged cave on the south wall of the antechamber of Cave 209. It is not included in the current numbering system. |
| 7 | 103/(104+105+103a) | Antechamber, south and north walls | Early Tang+(mid- through late Tang); | Caves 104 and 105 are buddha image shrines with sculpted canopy-shaped niches typical of the mid-Tang period, whereas Cave 103a was a shadow cave of which a monk statue was recorded in the early twentieth century but is no longer extant. |
| 8 | 323/(324+325) | Antechamber, south and north walls | Early Tang+(Xixia+Five Dynasties), Five Dynasties and Xixia renovations | Two ear caves added, respectively, during two renovations. |
| 9 | 335/(336+337) | Corridor north wall and antechamber south wall | Early Tang+(late Tang), mid-Tang and Yuan renovations | The two ear caves were added later, probably during or between the subsequent renovations of the main cave. |

**Table A1.** *Cont.*

| Set No. [1] | Cave Nos. [Main Cave(s)/Auxiliary Cave(s)] | Location of the Ear Caves or Auxiliary Caves [2] | Construction Periods [Main Cave(s)+(Auxiliary Cave(s)], Renovation Periods of the Main Cave | Brief Description of the Formation of the Cave Suite and Special Function of the Ear Cave If Applicable |
|---|---|---|---|---|
| 10 | 342/343 | Corridor, north wall | Early Tang+(late Tang), Five Dynasties renovation | Cave 343 was originally cut onto the north wall of the corridor of Cave 342, and it was concealed in the Five Dynasties period. At some point later, it was broken into from the east wall of the main chamber of Cave 342. |
| 11 | 347/(348+349) | Corridor, south and north walls | High Tang+(late Tang), Five Dynasties and Xixia renovation | The two ear caves were added later than the main chamber, probably during a renovation of the main cave, and the cave suite was together refurbished in the Xixia period. |
| 12 | 225/(226+227) | Antechamber, west wall | High Tang+(mid-Tang+late Tang), mid-Tang and Five Dynasties renovations; | Cave 226 was added during the first time of renovation, whereas Cave 227 was probably added during the second time of renovation. |
| 13 | 166/(167+168) | Antechamber, west wall | High Tang+(late Tang), mid-Tang, Five Dynasties, Song renovations | The ear caves were added later, and Cave 167 and the main cave were renovated in the same period (Song). |
| 14 | 182/(181+183) | Antechamber, west wall | High Tang+(late Tang), Song renovation | The ear caves were added later. |
| 15 | 186/(187) | Antechamber, west wall south | Mid-Tang+(Five Dynasties), Five Dynasties renovation | The two caves share an antechamber. |
| 16 | 172/173 | Corridor, south wall | High Tang+(late Tang), Song renovation | The ear cave was added during a renovation of the main cave. |
| 17 | 176/(177+178) | Corridor, south and north walls | High Tang+(late Tang); mid-Tang and Song renovations | The two ear caves were added later, and the cave suite together was renovated in the Song period. |
| 18 | 175+174/(174 niche) | Antechamber, north wall | High Tang+(Song), Song renovation | The niche on the north wall of Cave 174 (i.e., the antechamber of Cave 175) functioned as a shadow cave. |
| 19 | 23/24 | Antechamber, south wall | High Tang+(late Tang), mid-Tang and Five Dynasties renovations | The ear cave was added during or in-between the subsequent renovations of the main cave. |
| 20 | 188/189 | Antechamber, north wall | High and mid-Tang+(Five Dynasties), Five Dynasties, and Song renovations | The ear cave was added and renovated during the subsequent renovations of the main cave, whose construction was initiated in High Tang and completed in the mid-Tang period. |
| 21 | 194/195 | Antechamber, north wall | High Tang+(late Tang), late Tang and Xixia renovations | The auxiliary cave was added during the first renovation of the main cave. |
| 22 | 197/(191+190) | Antechamber, south wall | Mid-Tang+(mid-Tang+late Tang), Five Dynasties and Song renovations | Caves 197 and 191 were constructed in the same period, whereas Cave 190 was added later. |

Table A1. *Cont.*

| Set No. [1] | Cave Nos. [Main Cave(s)/Auxiliary Cave(s)] | Location of the Ear Caves or Auxiliary Caves [2] | Construction Periods [Main Cave(s)+(Auxiliary Cave(s)], Renovation Periods of the Main Cave | Brief Description of the Formation of the Cave Suite and Special Function of the Ear Cave If Applicable |
|---|---|---|---|---|
| 23 | 130/(130a+130b) | Corridor, south and north walls | High Tang+(High Tang), Song renovation | Caves 130a and 130b are located on the upper part of the ground-level corridor of Cave 130, and their positions indicate the existence of a mezzanine level in the ante-hall of Cave 130. |
| 24 | 74/73 | Antechamber, north wall | High Tang+(Song), Five Dynasties renovation | The ear cave was added later, probably during the renovation of the main chamber. At some point later, Cave 73 was broken into from the antechamber of Cave 72. |
| 25 | 134/(133+135) | Antechamber, south and north walls | Mid-Tang+(mid-Tang), late Tang and Song renovations | The three caves were made and renovated around the same periods. |
| 26 | (376+378)/377 | Antechamber, west wall middle | Sui+(Song); Song renovation; | The ear cave was added during the renovation of the two main caves that were adapted to share an antechamber. |
| 27 | 386/385 | Antechamber, west wall north | Early Tang+(Five Dynasties); mid-Tang, Five Dynasties renovations | The ear cave was added later around the Five Dynasties renovation of the main cave, which seems to be contingently constructed from the early to mid-Tang periods. |
| 28 | 358/357 | Cliff face, above the entrance corridor | Mid-Tang+(mid-Tang), Five dynasties, Xixia renovation | Cave 357 enshrines a meditating monk's statue and therefore might have been adapted to serve as a shadow cave. It was likely sheltered under a roof during a renovation of the main cave. |
| 29 | 53/(469+52) | Main chamber and corridor, north walls | Mid-Tang+(mid-Tang+High Tang), Five Dynasties renovation | Cave 53 was expanded in the Five Dynasties, and Cave 469 was used as a sūtra storage with an inscription dated 953 CE. |
| 30 | 158/(157+158a+158b) | Cliff face | Mid-Tang+(mid-Tang+unknown period), Xixia renovation | Caves 157 and 158b seem to be a pair of ear caves which were made around the same time with the main cave, whereas Cave 158a seems unfinished. |
| 31 | 159/(160) | Antechamber, north wall | Mid-Tang+(late-Tang) | The ear cave was added to the main cave and was later broken into from the cliff. |
| 32 | 29/(490+28) | Cliff face | Late Tang (+High Tang) | Cave 29 is inserted in between the pre-existing Caves 490 and 28 or expanded from another pre-existing cave, and Cave 29's antechamber partly destroyed Caves 490 and 28. |

Table A1. *Cont.*

| Set No. [1] | Cave Nos. [Main Cave(s)/Auxiliary Cave(s)] | Location of the Ear Caves or Auxiliary Caves [2] | Construction Periods [Main Cave(s)+(Auxiliary Cave(s)], Renovation Periods of the Main Cave | Brief Description of the Formation of the Cave Suite and Special Function of the Ear Cave If Applicable |
|---|---|---|---|---|
| 33 | 16/17 | Corridor, north wall | Late Tang+(late Tang), Song/Xixia renovation | The ear cave was made at the same time or slightly later than the main cave and then concealed during the renovation of the latter. Cave 17 successively served as a shadow cave and for storage. |
| 34 | 12/(11+13) | Antechamber, north and south walls | Late Tang+(late Tang), Five dynasties renovation | The ear chambers were likely made at the same time as the main cave, and Cave 11, which was fully refurbished in the Qing, exhibits positional and typological features of the late-Tang shadow caves. |
| 35 | 9/(8+10) | Antechamber, north and south walls | Late Tang+(late-Tang), Song and Yuan renovations | The main cave, auxiliary cave, and ear cave were made around the same time. |
| 36 | 136/137 | Antechamber, north wall | Late Tang+(Five Dynasties), Song and Xixia renovation | The ear cave was made at the same time or slightly later than the main cave and then half destroyed by an expansion of the antechamber of the latter. Cave 137 served as a shadow cave. |
| 37 | 138/139 | Antechamber, north wall | Late Tang+(late Tang), Five Dynasties and Yuan renovations | The ante-hall of Cave 138 destroyed the ground and walls of a pre-existing cave; the main cave and the ear cave were made at the same time. Cave 139 served as a shadow cave. |
| 38 | 152/(153+154) | Antechamber, west wall | Song+(mid-Tang), Uighur/Xixia renovation | The main cave was inserted between two pre-existing caves and the former's antechamber partly destroyed one of the latter (Cave 154). The cave suite together was renovated later. |
| 39 | 261/261a | Antechamber, south wall | Five Dynasties+(Five Dynasties) | The main cave and the ear cave were made at the same time. |
| 40 | 22/22a | Corridor, south wall | Five Dynasties+(Five Dynasties) | The main cave and the ear cave were made at the same time. |
| 41 | 55/(55a+56+478+two unnumbered caves) | Antechamber, west wall | Song+(Sui+mid-Tang); Song renovation | The corridor of Cave 55 was located right below Sui Cave 55a, and the ante-hall of Cave 55 leads to the destruction and concealment of Sui Cave 56 and mid-Tang Cave 478 and two un-numbered caves. |
| 42 | 444/443 | Antechamber, north wall | High Tang+(Song), Song renovation | The ear cave was made during the renovation of the main chamber. Cave 443 served as a shadow cave. |

[1] The table follows a chronological order: the initial construction period of the earliest cave in the cave group. The periods mentioned include the Northern Liang (398–439 CE), the Northern Wei (439–535), the Western Wei (535–557), the Northern Zhou (557–581), the Sui (581–618), the early Tang (618–710), the High Tang (710–781), the mid-Tang (781–850, also known as the Tibetan (Tubo) period), the late Tang (851–907), the Five Dynasties (907–960), the [early Northern] Song (960–1036), the Tangut (Xixia) period (1036–1227), and the Yuan period (1271–1368). [2] In this study, "ear cave" refers to inhabitable caves and niches, whereas "auxiliary cave" refers to caves ample enough to be entered.

## Notes

1   As Mahāyāna Buddhism, the form of Buddhism in East Asia, embraces the concept of myriad Buddhas, it also develops the Buddha lands of ten directions and three times. In addition to the Western Pure Land, the popular Pure Lands include the Eastern Pure Land of Bhaiṣajyaguru, the medicine Buddha, and the Pure Land of Maitreya, the future Buddha.

2   Western Pure Land tableaux in high-Tang Dunhuang caves include 20 Meditation Sūtra tableaux in Mogao caves 45, 66, 103, 113, 116, 120, 122, 148, 171 (3 pieces), 172 (2 pieces), 176, 194, 208, 215, 217, 320, and 446, and 5 Amitābha/Amitāyus Sūtra tableaux in Mogao Caves 44, 205, 445, 66, and 225.

3   There is much scholarship on *bianxiang*. See, for example. Wu (1992b).

4   Discussions of this Pure Land transformation tableaux on the north wall of Mogao Cave 172 are numerous; for a recent literature review, see Feng (2018, pp. 2–29). Architectural historian Xiao Mo considers this painting to be a quintessential example of architectural painting in ancient China. Xiao, Sun Ruxian, and Sun Yihua, among others, have proposed various layouts of the courtyard complex it depicts. Art historian Wu Hung instead examines the visual modes of the tripartite picture. See Xiao (2019, pp. 76–79, 293–97); Sun and Sun (2001, pp. 116–17, 125–27); Wu (1992a).

5   An *aspirant* refers to a sentient being who is born in the blissful land of Amitābha by transformation. The age of an aspirants at the moment of the transformational rebirth is not specified in Buddhist scriptures, but medieval Chinese painters usually depict them as babies.

6   While the most general practice is "being mindful of the (Amitābha) Buddha", a specific meditation process called the "sixteen meditations" is introduced in the *Meditation Sūtra*. A historical discussion is included in a work of the eminent Tang monk Shandao 善導's (613–81), *Guannian Amitofo xianghai sanmei gongde famen* 觀念阿彌陀佛相海三昧功德法門 [Methods for the merit of samādhi by visualizing the sea-like Image of Amitāyus-Amitābha].

7   A *kalpa* (*jie* 劫) or aeon is the immensely long period of time defining the cycle of creation and re-creation of the universe in Buddhist and Hindu cosmology.

8   In general speaking, visualization is an important means of practice in Mahāyāna Buddhist theories. A practitioner, usually assisted by visual objects, actively evokes in their mind the presence of the divine beings and their dwelling place, for the purposes of self-identification and transformation. For recent studies on Buddhist visualization techniques, see Greene (2016) and Anderl (2020).

9   "歸去來，寶門開，正見彌陀升寶座，菩薩散花稱善哉，稱善哉。"    "歸去來，見彌陀，今在西方現說法，拔脫眾生出愛河，出愛河。"

10  "一一觀之作七重行樹想。……一一樹上有七重網。一一網間有五百億妙華宮殿。" "極樂國土。七重欄楯七重羅網七重行樹。皆是四寶周匝圍繞。是故彼國名曰極樂。"

11  Chinese monks Huiyuan of the Jingying Temple 淨影慧遠 (523–92), Zhiyi 智顗 (538–79), Jizang 吉藏 (549–623), and Shandao offered four kinds of categorization of the sixteen meditations. This study follows Shandao's categorization.

12  For the sake of comparison, the technique of multiple vanishing points is also found in ancient Roman art wall painting (Little 1971), although the diagonal lines do not necessarily parallel one another like in the Mogao Cave 172 Pure Land painting. Art historians invented the term "vanishing vertical axis" "fishbone" and "herringbone" to describe a central vertical axis in a painting with multiple points for the placing of diagonal lines that shows the recession of visual elements in a spatial setting (Panofsky 1991, pp. 102–5). Yet the theory of vanishing vertical axis applied in the Roman context has been contested by a recent study by Small (2019), which suggested that concentric circles, grids, or a combination were more likely used for the design of Roman painted walls. The current study investigates, rather than painting techniques, the religious meaning the perspective potentially convey.

13  Excerpt from the Wu Sengtong stele (P. 4640), the most detailed accounts of the biography and cave construction activities of a Dunhuang monk official Wu Hongbian (d. 862). The manuscript is a copy around 900 CE of a commemorative stele whose text was compiled by a local scholar Dou Liangji in around 834 CE.

14  The dating of the ear-chamber Cave 173, as well as other caves in this study, is after Dunhuang Academy's dating.

15  While *plastic* denotes "sculptural or pliable", my use of the term follows the American architect Frank Llyod Wright's use: "light and continuously flowing instead of the heavy 'cut and butt' of the usual carpenter work."

16  Data is based on measurements in the digital 3D model. For a discussion of the visuality of such design and a potential context of artistic competition, see Wu (1992a, pp. 59–60).

17  For a discussion of the visual and bodily experience of architecture, see Pallasmaa (2005, pp. 6–80).

18  Previously, Zhao and Duan (2019, pp. 165–67) surveyed archaeological drawings made by Shi Zhangru and concluded that the Mogao complex contains 69 cave groups in total. The current study, which is based on in situ fieldwork and records of early international expeditions, identifies some cases that were not represented in Shi (1996) or recognized by Zhao and Duan (2019) and removes those that do not meet the authors' criteria for a "cave group".

19  Among them, there are two types of miniature shrines. The first type, comprising of six cases, are memorial cave chapels of eminent monks. The second type, accounting for the rest, are caves or niches enshrining images of Buddhist deities. Neil Schmid,

researcher at the Dunhuang Academy, is one of very few scholars currently investigating these types of caves. Scholarship on the topic is otherwise scarce.

20 Excerpt from Zhang Yingrun's inscription on the antechamber wall of Cave 108 dated 939 CE.

21 Many Dunhuang scholars believe that the Cao-family Guiyijun period, ca. 914–1036, saw a massive construction of timber façades, ante-halls, and open-air murals, although a visual analysis has seldom been attempted.

22 Unfortunately, the pictorial contents of the mural cannot be discerned from the monochromic photo or from the little that remains of the mural today.

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
