# Peer review of "Vision and Site: Revisiting a Pure Land Cave of Dunhuang"

_religions, doi:10.3390/rel15030329_

Round 1

Reviewer 1 Report

Comments and Suggestions for Authors

This is a superior article.  Although there are countless studies of Dunhuang murals and of Pure Land imagery, this article has an original approach.

Comments on the Quality of English Language

The English is fine.  Article should have standard editing.

Author Response

Thank you for the review!

Reviewer 2 Report

Comments and Suggestions for Authors

70 Amitābha  not “Amitabha”, since the author seems to apply the Sanskrit diacritics, this has to be consistent throuhgout the article 

76 as the later paintings did not “as the later paintings do”

81 Skt. Bhaiṣajyaguru not “Bhaişajyaguru” Wrong diacritics in “s”

134 apsara is a Sanskrit word but apsaras is not, I suggest putting the Sanskrit in italics to distinguish it from the English addition of the plural, i.e. apsaras

146 the same, it should be kalpas not “kalpas’

164 Amitābha not “Amitabha”

203 Some references on the perspective theory in premodern Chinese contexts should be supplied here, secondary literature, etc. 

212 one-point perspective in post-Reinaissance art, please supply some references. 

257 Amitāyus not “Amitayus”

258 Mahāsthāmaprāpta not “Mahasthamaprapta”. The consistency of applying Sanskrit diacritics is not maintained.

262 “it was extensively discussed”, what is “it”, please provide the object of the sentence

264 samādhi, not “samadhi”, Amitāyus-Amitābha not Amitayus-Amitabha

268 “fix topics”or five topics? If we count 7 + 5+ 3 it gives 15 not 16. Perhaps some explanation of the number discrepancy is needed in the footnote

284-289 is this multiple vanishing point also present in other architecture theories? Is it unique for this period? Perhaps a note on that is needed, since the author has already referred to the post-Reinaissance art. 

291-297, the part of visualization would benefit by expanding it a little bit, and discussing a Buddhist theory of visualization, see the reference in Oxford encyclopedia “Visualization is A form of meditation where an image of a Buddha or some other divine being is creatively imagined for purposes of devotion or spiritual transformation. Pre-Mahāyāna Buddhist forms of visualization include the use of kasiṇas and other inanimate objects, while in early Mahāyāna such methods are taught in texts such as the Sukhāvatī-vyūha Sūtra and the Pratyutpanna Sūtra in connection with evoking the presence of an idealized form of a Buddha and his dwelling place. Visualization was further developed to form the cornerstone of tantric practice (sādhana) in which an individual creates an image of a divine being for the purposes of self-identification and transformation”.

In other words, something on the visualization as a religious practice. A lot of literature has been written on Buddhist visualization techniques in recent years, for example, 

VISIONS AND VISUALIZATIONS: IN FIFTH-CENTURY CHINESE BUDDHISM AND NINETEENTHCENTURY EXPERIMENTAL PSYCHOLOGY

EricM. Greene

History of Religions

Christoph Anderl, Some References to Visualization Practices in Early Chán Buddhism with an Emphasis on guān  and kàn EMAIL logo

From the journal Asiatische Studien - Études Asiatiques

640 Amitābha

643 Amitāyus-Amitābha

Author Response

Thank you very much for taking the time to review this manuscript. Please find the detailed responses below and the corresponding revisions/corrections in track changes in the re-submitted files.

  1. All the Sanskrit words were corrected or put in italics according to your comments. Thanks very much for pointing out for me.
  2. As for the comment “203 Some references on the perspective theory in premodern Chinese contexts should be supplied here, secondary literature, etc.” I adapted note 10 to main text in last paraph on page 7, lines 245, 255-258.
  3. As for the comment “212 one-point perspective in post-Reinaissance art, please supply some references.” I added two references (Panofsky 2002: 27–39, Gioseffi 1967) in line 261.
  4. As for the comment “268 “fix topics”or five topics? If we count 7 + 5+ 3 it gives 15 not 16. Perhaps some explanation of the number discrepancy is needed in the footnote”, I found “fix” a typo of “six” and corrected it.
  5. As for the comment “284-289 is this multiple vanishing point also present in other architecture theories? Is it unique for this period? Perhaps a note on that is needed, since the author has already referred to the post-Reinaissance art.” I added a note about similar perspectival constructions in ancient Roman wall paintings. Note 12, lines 708-715.
  6. As for the comment “291-297, the part of visualization would benefit by expanding it a little bit, and discussing a Buddhist theory of visualization…” I discussed the most relevant theory and practice of visualization in lines 165-192 and supplied the general ideas and secondary literature you recommended in note 8.

Reviewer 3 Report

Comments and Suggestions for Authors

The author meticulously depicts the details and intricacies of the mural painting and the overall architectural and visual designs in Cave 172 in Dunhuang, coupled with a variety of simulation techniques. This renders the article with an in-depth analysis from some very innovative focal points. I am particularly fascinated by the detailed description of the sequential viewing of the sixteen meditations and the virtual walk-through experience. These imageries allow the reader to acquire an in-situ viewing experience and efficiently convey the central argument regarding the utility of these mural paintings for Buddhist pilgrims in medieval China. In general, I find this article well articulated and worthy of publication.

There are several suggestions I would like to make. First, the author directly goes into the discussion of Cave 172, merely mentioning it as "one of the representative Pure-Land caves" (Line 36). A bit more discussion on the grand scale about Pure Land art in Dunhuang, particularly its distribution among these caves, and the reason why the author wishes to focus on this cave, would be helpful for readers to understand the importance and contribution of the article to the study of Dunhuang caves, Pure Land arts, and Buddhism in medieval China.

Second, the argument about the relation between Pure Land imagery and the architectural space of the cave and cave suite is well presented in the article. However, I wonder if these imageries have any impact on Pure Land practitioners in medieval China. For many Pure Land practitioners, visualization meditation is central to their religious cultivation. Therefore, is there any link between this Dunhuang visualization of the Pure Land Paradise and the daily practice of Pure Land followers? In other words, is there any real influence of this mural painting on Pure Land practitioners? This might be a little beyond the scope of this article, but it could make the central argument more related to the study of medieval Buddhism in China.

Lastly, I notice that the main discussion of the mural painting in Cave 172 is about the north wall painting, while there is also a south wall painting with similar themes composed at a different time. Is there any reason why the author would not want to discuss the painting on the south wall more?

Author Response

Thank you very much for taking the time to review this manuscript. I found your suggestions thoughtful and constructive and agreed with all of them. Please find the detailed responses below and the corresponding revisions/corrections in track changes in the re-submitted files.

  1. As for the comment “A bit more discussion on the grand scale about Pure Land art in Dunhuang, particularly its distribution among these caves, and the reason why the author wishes to focus on this cave…”, I added a brief overview of high-Tang pure land paintings and caves in the first paragraph of the introduction section, lines 32-37.

  1. Your second point is a very important and difficult question, which I always hope to discuss but could not give a complete answer. I tried to response by reviewing the relationship between the cave art, religious practices, and the shared visual culture displayed in ritual texts and image. See lines 165-192. Unfortunately, I could not determine whether this mural painting had any real influence on Pure Land practitioners in medieval Dunhuang, as no direct evidence is found. I will continue to consider this issue about Pure Land image and practice in future studies.

  1. As for your third point “Is there any reason why the author would not want to discuss the painting on the south wall more?”, I added a bit discussion and my reason of focusing on the north-wall painting in lines 120-142.